# Quantifying the impact of immune history and variant on SARS-CoV-2 viral kinetics and infection rebound: A retrospective cohort study

James A Hay[1†], Stephen M Kissler[1†], Joseph R Fauver[2,3†], Christina Mack[4†], Caroline G Tai[4], Radhika M Samant[4], Sarah Connolly[4], Deverick J Anderson[5], Gaurav Khullar[6], Matthew MacKay[6], Miral Patel[6], Shannan Kelly[6], April Manhertz[6], Isaac Eiter[6], Daisy Salgado[6], Tim Baker[6], Ben Howard[6], Joel T Dudley[6], Christopher E Mason[6], Manoj Nair[7], Yaoxing Huang[7], John DiFiori[8,9], David D Ho[7], Nathan D Grubaugh[2*], Yonatan H Grad[1*]

[1]Harvard TH Chan School of Public Health, Boston, United States; [2]Yale School of Public Health, New Haven, United States; [3]University of Nebraska Medical Center, Omaha, United States; [4]IQVIA, Real World Solutions, Durham, United States; [5]Duke Center for Antimicrobial Stewardship and Infection Prevention, Durham, United States; [6]Tempus Labs, Chicago, United States; [7]Vagelos College of Physicians and Surgeons, Columbia University, New York, United States; [8]Hospital for Special Surgery, New York, United States; [9]National Basketball Association, New York, United States

*For correspondence:
grubaughlab@gmail.com (NDG);
ygrad@hsph.harvard.edu (YHG)

†These authors contributed equally to this work

## Abstract

**Background:** The combined impact of immunity and SARS-CoV-2 variants on viral kinetics during infections has been unclear.

**Methods:** We characterized 1,280 infections from the National Basketball Association occupational health cohort identified between June 2020 and January 2022 using serial RT-qPCR testing. Logistic regression and semi-mechanistic viral RNA kinetics models were used to quantify the effect of age, variant, symptom status, infection history, vaccination status and antibody titer to the founder SARS-CoV-2 strain on the duration of potential infectiousness and overall viral kinetics. The frequency of viral rebounds was quantified under multiple cycle threshold (Ct) value-based definitions.

**Results:** Among individuals detected partway through their infection, 51.0% (95% credible interval [CrI]: 48.3–53.6%) remained potentially infectious (Ct <30) 5 days post detection, with small differences across variants and vaccination status. Only seven viral rebounds (0.7%; N=999) were observed, with rebound defined as 3+days with Ct <30 following an initial clearance of 3+days with Ct ≥30. High antibody titers against the founder SARS-CoV-2 strain predicted lower peak viral loads and shorter durations of infection. Among Omicron BA.1 infections, boosted individuals had lower pre-booster antibody titers and longer clearance times than non-boosted individuals.

**Conclusions:** SARS-CoV-2 viral kinetics are partly determined by immunity and variant but dominated by individual-level variation. Since booster vaccination protects against infection, longer clearance times for BA.1-infected, boosted individuals may reflect a less effective immune response, more common in older individuals, that increases infection risk and reduces viral RNA clearance rate. The shifting landscape of viral kinetics underscores the need for continued monitoring to optimize isolation policies and to contextualize the health impacts of therapeutics and vaccines.

**Funding:** Supported in part by CDC contract #200-2016-91779, a sponsored research agreement to Yale University from the National Basketball Association contract #21-003529, and the National Basketball Players Association.

### Editor's evaluation

This manuscript provides a valuable and policy-relevant contribution to our understanding of SARS-CoV-2 viral kinetics in the Omicron era. The authors exploit a rich and unique dataset from the National Basketball Association to describe post-infection viral kinetics, including rebounds, and to explore evidence for differential kinetics by immune history and demographics. The authors show (as others have) that most people remain with high viral loads 5 days post positive test and that older individuals and those who were boosted (but had a poor initial antibody response to the primary vaccine series) were more likely to remain with high viral loads longer after an Omicron infection.

## Introduction

The viral kinetics of SARS-CoV-2 underlie the epidemiology of COVID-19 and the policies surrounding infection control. The amount and duration of viral shedding influences infectiousness (*Ke et al., 2021*; *Ke et al., 2022*; *Marc et al., 2021*; *Marks et al., 2021*; *Puhach et al., 2022*; *Sun et al., 2021*) and the duration of test positivity affect isolation policies, test recommendations, and clinical care guidelines (*Hellewell et al., 2021*; *Kissler et al., 2021a*; *Larremore et al., 2021*; *Mack et al., 2022*; *Néant et al., 2021*; *Quilty et al., 2021*; *Singanayagam et al., 2022*). Descriptions of viral kinetics are also important for establishing baselines to measure the effectiveness of antiviral drugs. For example, rebounds of viral RNA concentrations and symptoms have been observed after antiviral treatment, but it has been unclear to what extent such rebounds also occur in the absence of drug (*Boucau et al., 2022b*; *Charness et al., 2022*). Most longitudinal viral kinetics studies pre-date the emergence of the Omicron variant, which features dramatic antigenic divergence from prior variants, as well as the rollout of third and fourth vaccine doses (*Lusvarghi et al., 2022*; *van der Straten et al., 2022*). Early findings on viral kinetics therefore need to be updated to account for extensive and heterogeneous immune experience across the population (*Cevik et al., 2021*; *Kissler et al., 2021b*).

To characterize the viral kinetics of SARS-CoV-2 infection, including rebounds, for the Delta and Omicron (BA.1 lineages BA.1.1529 and BA.1.1) variants in symptomatic and asymptomatic individuals with varied vaccination and infection histories, we measured viral RNA levels using densely-sampled RT-qPCR tests from 1,280 SARS-CoV-2 infections, each taken by combined anterior nares and oral swabs, that occurred between 7th July, 2020, and 26th January, 2022, prior to the detection of BA.2.12.1, BA.4 and BA.5, or the regular detection of BA.2, in this cohort. As a proxy for immune response to SARS-CoV-2, we used antibody titers against the ancestral SARS-CoV-2 (WA1) strain spike protein measured prior to the administration of booster doses, but predominantly after primary vaccination.

We interpreted the data in two ways. First, we estimated the probability of an individual having a PCR cycle threshold (Ct) value less than 30, as a proxy for infectiousness, on each day post detection using a logistic regression model. Second, we estimated the peak viral RNA concentrations, viral RNA proliferation rate, and viral RNA clearance rate across variants, immune statuses and age using a semi-mechanistic model. Our findings provide key estimates for the duration and magnitude of viral RNA shedding in the upper respiratory tract and its variation across age, symptom status, variants, immune states, and individuals.

## Methods
### Study design

The data reported here represent a convenience sample including team staff, players, arena staff, vendors, and others affiliated with the NBA as described previously (*Kissler et al., 2021a*; *Kissler et al., 2021b*). The retrospective study includes samples collected between 7th July 2020 and 26th January 2022 (*Appendix 1—figure 1*). Clinical samples were obtained by combined swabs of the

anterior nares and oropharynx, collected separately from each anatomical site, for each patient administered by a trained provider. Daily testing was required for most individuals prior to vaccination availability, with less frequent testing but close monitoring required after vaccination. Cycle threshold (Ct) values were generated using the Roche cobas target 1 assay. For the viral kinetics model analyses, Ct values were converted to viral genome equivalents using a standard curve (*Kissler et al., 2021a*).

We classified all individuals as having Ct value <30 or not on each day post-detection. This threshold was chosen based on a combination of antigen sensitivity and studies of virus culture by Ct, where the presence of culturable virus is often assumed to correlate with infectivity (*Brihn et al., 2021*; *Bullard et al., 2020*; *Pilarowski et al., 2021*; *Singanayagam et al., 2020*; *Thommes et al., 2021*). We stratified infections by those who had a negative or inconclusive test ≤1 day prior to detection and those whose last negative or inconclusive test was ≥2 days ago. We assumed that individuals testing negative at the end of an acute infection remained negative for the remainder of the study period, whereas those ending in a positive test are right-censored. Rebound trajectories were defined as any trajectory with a sequence of two or more consecutive Ct values ≥30 or negative tests after the initial peak followed by two or more consecutive Ct values <30. We considered more stringent definitions both for initial clearance (3+ or 4+ days of Ct ≥30 or negative test following initial peak) and subsequent rebound (3+ or 4+ days of Ct <30). In some instances, individuals were tested multiple times per day and thus for ease of model fitting we excluded 3,751 positive or inconclusive and 14,713 negative samples from repeat tests on the same day in our analyses, prioritizing the earliest test and then lowest Ct value test on each day.

Vaccination information was reported and verified by NBA staff and a clinical operational team. 828 individuals had been boosted by the time of their last PCR test, 529 had completed their primary vaccination course (two doses of an mRNA vaccine or one dose of Janssen / Ad.26.COV2.S adenovirus vector-based vaccine), 8 had received one vaccine dose, and 13 confirmed to be unvaccinated. The vaccination statuses of the remaining individuals were unknown. The time course of individual vaccination and exposure times is shown in *Appendix 1—figure 2*.

## Study oversight

In accordance with the guidelines of the Yale Human Investigations Committee, this work with de-identified samples was approved for research not involving human subjects by the Yale Institutional Review Board (HIC protocol # 2000028599). This project was designated exempt by the Harvard Institutional Review Board (IRB20-1407).

## Classification of infections

We tagged each series of positive tests buffered by at least 14 days of negative or missing tests on each side as a distinct infection. After an infection was flagged, subsequent positives were not classified as a new infection for 90 days. Isolated positive tests with no other positive within 14 days either side were not considered as detections. We track the cumulative number of exposures (defined as either receiving a vaccination or infection) over time. Individuals who received the Janssen/Ad.26. COV2.S adenovirus vector-based vaccine were counted as having received two vaccine doses. A total of 351 additional infections were reported to the program outside of the main testing regime, either through an external PCR or rapid antigen test, or from a positive antibody test result (not including the Diasorin Trimeric Assay results described below). We consider these detections as contributing towards an individual's infection history but are unable to include them in the Ct value trajectory analyses.

## Genome sequencing and lineage assignment

RNA was extracted and confirmed as SARS-CoV-2 positive by RT-qPCR (*Vogels et al., 2021*). Next Generation Sequencing was performed with the Illumina COVIDSeq ARTIC viral amplification primer set (V4, 384 samples, cat# 20065135). Library preparation was performed using the amplicon-based Illumina COVIDseq Test v033 and sequenced 2×74 on Illumina NextSeq 550 following the protocol as described in Illumina's documentation. The resulting FASTQs were processed and analyzed on Illumina BaseSpace Labs using the Illumina DRAGEN COVID Lineage Application; (*BaseSpace Labs, 2021*) versions included are 3.5.0, 3.5.1, 3.5.2, 3.5.3, and 3.5.4. The DRAGEN COVID Lineage pipeline was run with default parameters recommended by Illumina. Lineage assignment and phylogenetics

analysis using the most updated version of Pangolin (*Rambaut et al., 2020*) and NextClade (*Aksamentov and Neher, 2021*), respectively. All sequenced Omicron infections were lineage BA.1 apart from 1 BA.2.10 infection. Sequenced Delta infections were a combination of lineages B.1.617.2 and AY.x.

There were 3 and 482 non-sequenced infections in the window of time when Alpha was replaced by Delta (29th May 2021 to 18th July 2021) and after the first detection of Omicron BA.1 (3rd December 2021 onwards), respectively (*Appendix 1—figure 3*). We removed these 485 infections from variant-specific analyses and assigned all non-sequenced infections prior to the detection of Omicron BA.1 to the dominant lineage at the time of detection (i.e. all infections prior to 29th May 2021 were assumed 'Other' and all infections between 18th July 2021 and 3rd December 2021 were assumed 'Delta'). We removed all non-sequenced infections detected after 3rd December 2021 from variant-specific analyses rather than classifying them as Omicron BA.1 due to the continued presence of Delta. Omicron BA.2 was not regularly detected until after this period, with only one confirmed BA.2 infection (BA.2.10), which was removed from these analyses.

## Antibody titers

Individuals were tested with the Diasorin Trimeric Assay for IgG antibody titers against the ancestral SARS-CoV-2 (WA1) strain spike protein during the 2021 pre-season period (September-October 2021). The majority (>90%) of blood draws were from mid-September to early October 2021. We classified individuals with a titer of >250 AU/ml as being in the high titer group and in the low titer group otherwise, chosen based on its correlation with authentic virus neutralization results for wild-type and Delta (*Liu et al., 2021*; *Wang et al., 2021*). Specifically, an authentic virus neutralization titer of 100 was found to be well correlated with a 50% protective neutralization level for wildtype (*Khoury et al., 2021*) and found to correspond to a DiaSorin AU of 189.09 (95%CI: 147.61–235.75) (*Appendix 1—figure 4*). The cutoff of 250 was therefore chosen as a conservative upper bound classifying an individual as at lower risk of infection with Delta or wildtype SARS-CoV-2. Note that this cutoff does not predict infection risk with Omicron and was simply chosen as a proxy for an individual's immune competence.

## Logistic regression models

We used the RStan package *brms* to fit Bayesian logistic regression models estimating the probability of having Ct value <30 on each day post detection, fitting all models to the frequent testing and delayed detection datasets separately (*Bürkner, 2022*). As a baseline, we considered a model without variant-specific effects, using smoothing splines to estimate the probability of having a Ct value less than 30 on each day post detection. We then fitted additional logistic regression models, adding additional spline terms and intercepts for the category-specific effect of age group (<30 years, 30–50 years and >50 years), vaccination status, cumulative number of previous exposures, days since previous exposure (categorized as naive, <1 month, 1–3 months and >3 months), and/or variant with days since detection. In models including variant, we considered the interaction of variant with exposure history, vaccination status or days since exposure category. We did not add an interaction between age group and any other variable. All models were fitted to the frequent testing and delayed detection group datasets separately.

We ranked models based on the expected log predictive density and evaluated their classification accuracy and area under the receiver operator curve using *k*-folds cross-validation (25 folds). For the antibody titer analyses, we fitted Bayesian logistic regression models for the probability of Ct value <30 as a function of days since detection, stratified by the interaction of titer group (above or below 250 AU/ml), age group, variant and vaccination status. Further details on the fitting process can be found in the **Appendix 1**.

## Viral kinetic model

We extended a previously reported model for capturing SARS-CoV-2 viral kinetics to estimate the viral proliferation time, viral clearance time, and peak viral load by variant and immune status (*Kissler et al., 2021a*; *Kissler et al., 2021b*). The model approximates viral kinetics on a logarithmic scale as a piecewise linear function, corresponding to an exponential increase of virus followed by an exponential clearance at possibly different rates. To estimate the relationship between booster status and

viral kinetics, we first stratified the model by (1) Omicron boosted and (2) Omicron non-boosted individuals. There were too few boosted individuals who were infected with other variants to reliably fit the model to non-Omicron infections. Next, to estimate the relationship between antibody titer and viral kinetics, we stratified the model by (1) Delta infections with titer ≤250, (2) Delta infections with titer >250, (3) Omicron infections with titer ≤250 (4) Omicron infections with titer >250, and finally (5) non-Delta and non-Omicron infections in individuals who had not had any prior exposure either through infection or vaccination, to serve as a baseline. Stratification was accomplished by choosing a reference category (Omicron BA.1 non-boosted in the first analysis, non-Delta and non-Omicron infections without prior exposure in the second analysis) and fitting independent additive random effects for the other categories. Full details on the fitting procedure may be found in *Kissler et al., 2021a*; *Kissler et al., 2021b* and **Appendix 1**.

## Results
### Data

We initially identified 2,875 distinct infections from 2,678 individuals in this cohort (*Appendix 1—figure 1*). By the time of their final test, 2,460 (91.9%) individuals had one detected infection, 214 (7.99%) had two detected infections, three (0.11%) had three detected infections, and one (0.04%) had four detected infections. None of the individuals received antiviral treatment. A total of 587 infections were detected within 1 day of a prior negative PCR test result, and thus the timing of the onset of test positivity can be assumed with reasonable accuracy. We defined these infections as the 'frequent testing' group. The remaining 2,288 infections were detected 2 days or more from a previous negative test result or were detected with no prior negative test in the dataset. These were predominantly tests following suspected exposure, recent symptom onset, or periodic clearance for occupational health requirements, and thus we consider this latter group of detections as a reasonable proxy for infection detection in the absence of frequent testing, which is the case for most populations. We define these infections as the 'delayed detection' group.

Of 1086 infections with known symptom status, 766 reported symptoms at some point during the infection (70.5%). Individuals in the delayed detection group were more likely to be symptomatic than in the frequent testing group (73.1% vs 64.9%; Chi-squared test statistic = 5.03; p-value <0.05). Most symptomatic individuals were detected around the time of symptom onset (*Appendix 1—figure 5*; median delay from detection to symptom onset of zero days (N=553) in the delayed detection group and one day (N=171) in the frequent testing group). Symptom onset preceded the peak measured Ct

**Table 1.** Characteristics of the documented infections.
Counts (N) correspond to numbers of infections.

| Variable | Category | N | Percent |
|---|---|---|---|
| Total | – | 1280 | 100 |
| Variant | Delta | 180 | 14.1 |
| | Omicron BA.1 | 878 | 68.6 |
| | Other | 222 | 17.3 |
| Vaccination Status | Unvaccinated | 228 | 17.8 |
| | First dose | 6 | 0.5 |
| | Second dose | 420 | 32.8 |
| | Boosted | 626 | 48.9 |
| Antibody Titer | 13–250 | 473 | 37.0 |
| | 250–800 | 504 | 39.4 |
| | Unknown | 303 | 23.7 |
| Symptomatic | No | 257 | 20.1 |
| | Yes | 664 | 51.9 |
| | Unknown | 359 | 28 |
| Detection Speed | Delayed detection | 877 | 68.5 |
| | Frequent testing | 403 | 31.5 |
| Age Group | 0–30 | 556 | 43.4 |
| | 31–50 | 568 | 44.4 |
| | 50+ | 155 | 12.1 |
| | Unknown | 1 | 0.1 |
| Cumulative Infection Number | 1 | 1128 | 88.1 |
| | 2 | 149 | 11.6 |
| | 3 | 2 | 0.2 |
| | 4 | 1 | 0.1 |
| Days Since Previous Exposure | Naïve (no prior exposure) | 220 | 17.2 |
| | <1 month | 273 | 21.3 |
| | 1–3 months | 403 | 31.5 |
| | >3 months | 384 | 30.0 |

value by a median of two days (N=550) in the delayed detection group and three days (N=171) in the frequent testing group (*Appendix 1—figure 6*).

Based on genome sequencing, 1,561 infections were confirmed to be Omicron (one BA.2.10 isolate, the rest were lineages within BA.1), 266 confirmed to be Delta, and 247 confirmed as other lineages. An additional 801 infections were not sequenced; however, due to the rapid replacement of the circulating lineage in this cohort, we classified many of these as suspected Delta or other lineages based on the dominant variant at time of detection (*Appendix 1—figure 3*). We excluded non-sequenced samples following the detection of Omicron BA.1 due to the continued, albeit low-level, detection of Delta (N=490).

For further analysis, we reduced the dataset to a subset of 1,280 well-documented infections. Beginning with the 2,875 infections, we removed those with an unknown lineage (n=490) and one Omicron BA.2 infection, those with only binary test results (positive/negative but no Ct values; n=21), those for which all Ct-based tests results were beyond 25 days after the time of first detection (n=12), and those for which the vaccination status was missing (1,071). Characteristics of these infections are listed in *Table 1*.

## Interpersonal variation in viral RNA trajectories

Viral trajectories varied substantially across individuals regardless of variant (*Figure 1A*). To characterize the probability of an individual remaining potentially infectious on each day following detection, defined as having a Ct <30, we fitted a logistic regression model with a smoothing spline on days since detection as a predictor (more complex models are considered below). We fit this model to the frequent testing and delayed detection groups separately (*Appendix 1—figure 7*). Most individuals (posterior mean: 65.4%, 95% credible intervals [CrI]: 62.0–68.8%) in the frequent testing group remained potentially infectious on day 5 post detection. This fraction decreased to 20.0% (95% CrI: 17.3–22.8%) at day 10. In the delayed detection group, fewer individuals remained potentially infectious at days 5 and 10, likely because they were detected later in their infection. In this group, the proportion with Ct <30 was 51.0% (95% credible interval (CrI): 48.3–53.6%) on day 5 post detection and 9.37% (95% CrI: 7.98–10.9%) on day 10.

## Incidence of rebounds

We next characterized the frequency of rebound viral RNA trajectories in this cohort. Viral rebounds may be characterized by the duration of the "quiescent" period of low viral concentration between distinct peaks, the duration of the subsequent rebound, and the timing of rebound onset relative to infection, but no consensus definition of viral rebound based on these quantities exists. We defined rebound as any viral trajectory with a decline in Ct value to <30 for 3+consecutive days of tests (the rebound) after 3+consecutive days of tests with Ct ≥30 or a negative result (the quiescent period) following an initial Ct value <30 (the first detection of infection). Testing often ceased following initial clearance, and thus to minimize the impact of right censoring we only considered those trajectories with at least three days of tests with negative or Ct ≥30 following a Ct value <30 as the denominator (N=999). We detected seven viral rebounds under this definition. Less stringent definitions led to more rebound classifications. For example, 40 (3.00%) of 1,334 infections were identified as rebounds when only 2+consecutive days of Ct ≥30 followed by 2+days of Ct <30 was required to be classified as such (*Table 2*; *Figure 1B*). All individual-level viral trajectories classified as rebounds under this less stringent definition are shown in *Appendix 1—figure 8* Under this definition, we found that rebound infections were more likely in Omicron BA.1 infections, with 36 (4.10%; N=877) Omicron BA.1 infections resulting in rebound compared to one (0.562%; N=178) and three (1.08%; N=279) Delta and other infections, respectively (*Appendix 1—table 1*; Chi-squared test for Omicron BA.1 (N=877) vs. non-Omicron BA.1 infection (N=457), test statistic = 9.69, *P*-value <0.05). Similarly, we found that rebounds were more common in boosted individuals, with 32 (6.48%; N=494) rebounds in boosted individuals vs. three (0.929%; N=323) and two (1.26%; N=159) rebounds in vaccinated and unvaccinated individuals, respectively (*Appendix 1—table 2*; Chi-squared test for boosted (N=494) vs. not-boosted (N=478) infection, test statistic = 18.1, *P*-value <1e-4).

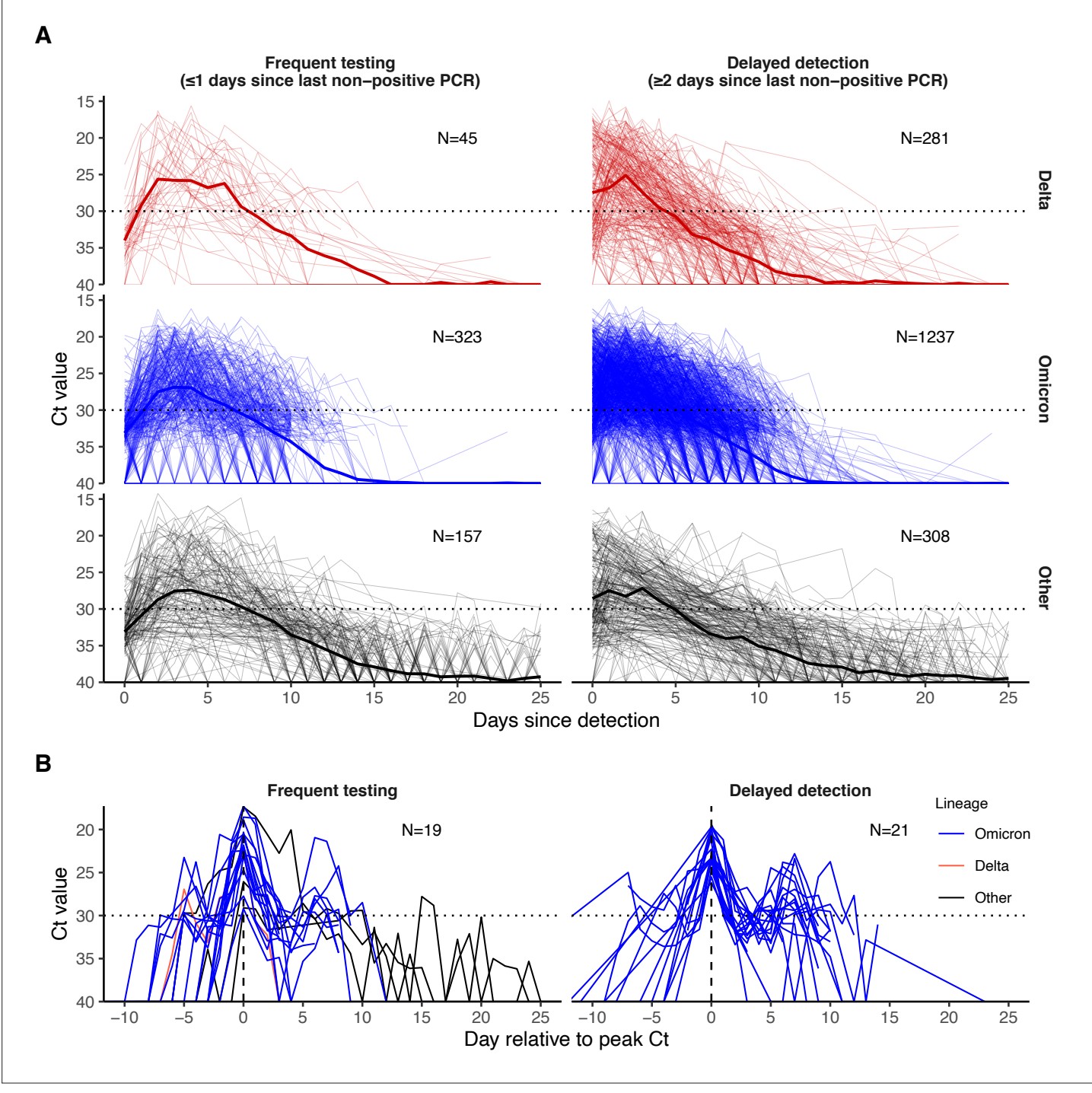

**Figure 1.** PCR Ct value trajectories for confirmed and suspected infections. (**A**) PCR Ct value trajectories for each acute Delta (red), Omicron BA.1 (blue), and other (black) infection. Individuals are grouped by the gap between detection and their most recent negative or inconclusive PCR test (Frequent testing vs. Delayed detection). Thick lines depict the mean Ct value over time, counting negative tests as Ct = 40. Thin lines depict individual level Ct values over time. The horizontal dotted lines mark Ct = 30, which we consider here as a proxy for possible infectiousness and antigen test positivity. (**B**) Subsets of PCR Ct value trajectories that were classified as rebounds, stratified by testing frequency group. Rebounds are defined here as any trajectory with an initial Ct value <30, followed by a sequence of two or more consecutive negative tests or tests with Ct value ≥30, and subsequently followed by two or more consecutive tests with Ct value <30.

**Table 2.** Number of rebound infections classified under different definitions for initial clearance and subsequent rebound.

| Initial clearance duration (consecutive days with Ct ≥30) | Rebound duration (days above Ct value threshold) | Ct value threshold of rebound | Rebounds | Total | Percentage |
|---|---|---|---|---|---|
| ≥4 | ≥4 | Ct <30 | 0 | 749 | 0.00% |
| ≥4 | ≥3 | Ct <30 | 1 | 749 | 0.13% |
| ≥4 | ≥2 | Ct <30 | 4 | 749 | 0.53% |
| ≥3 | ≥4 | Ct <30 | 2 | 999 | 0.20% |
| ≥3 | ≥3 | Ct <30 | 7 | 999 | 0.70% |
| ≥3 | ≥2 | Ct <30 | 16 | 999 | 1.60% |
| ≥2 | ≥4 | Ct <30 | 7 | 1334 | 0.53% |
| ≥2 | ≥3 | Ct <30 | 18 | 1334 | 1.35% |
| ≥2 | ≥2 | Ct <30 | 40 | 1334 | 3.00% |
| ≥4 | ≥4 | Ct <25 | 0 | 749 | 0.00% |
| ≥4 | ≥3 | Ct <25 | 0 | 749 | 0.00% |
| ≥4 | ≥2 | Ct <25 | 0 | 749 | 0.00% |
| ≥3 | ≥4 | Ct <25 | 1 | 999 | 0.10% |
| ≥3 | ≥3 | Ct <25 | 1 | 999 | 0.10% |
| ≥3 | ≥2 | Ct <25 | 2 | 999 | 0.20% |
| ≥2 | ≥4 | Ct <25 | 1 | 1334 | 0.08% |
| ≥2 | ≥3 | Ct <25 | 2 | 1334 | 0.15% |
| ≥2 | ≥2 | Ct <25 | 5 | 1334 | 0.38% |

## Minimal differences across variants and vaccination histories in the probability of having low Ct values over time

To assess differences in the duration of test positivity and infectiousness by age, variant and immune status, we modeled the probability of an individual having Ct <30 on each day since detection. As a baseline model, we fitted a logistic regression model with a smoothing spline on days since detection as a predictor. We then fitted successively more complex models, adding independent category-specific smoothing splines for the interaction of age group (categorized as <30, 30–50 or >50 years old), variant, and exposure history with days since detection (factorized), and compared their predictive accuracy using *k*-fold cross-validation. All models were fit to the frequent testing and delayed detection datasets separately.

The best-performing model for predicting the time course of low Ct values included days since detection stratified by the cumulative number of previous exposures (infection or vaccination) and its interaction with variant, as well as age group (*Appendix 1—table 3* and *Appendix 1—table 4*). This indicates that the variation in low Ct values over time is better captured by models that account for exposure history and age group than by models that account for time since detection alone. However, the models stratified by vaccination status or days since previous exposure in addition to variant and age group were also highly ranked, and the differences in classification accuracy among all of the models was small. The baseline model, which included only the number of days since detection as a predictor, gave an overall classification accuracy for an individual having Ct <30 or ≥30/negative of 81.7% with an AUC of 88.6% (group-level classification accuracies: Ct <30 = 60.6%; Ct ≥30/negative = 89.0%) for the frequent testing group, and an overall classification accuracy of 84.2% with an AUC of 90.5% (Ct <30 = 72.2%; Ct ≥30/negative = 88.0%) in the delayed detection group. In contrast, the best model, which included the cumulative number of exposures, variant and age, gave an overall classification accuracy of 82.8% with an AUC of 89.7% (Ct <30 = 64.7%; Ct ≥30/negative = 89.0%)

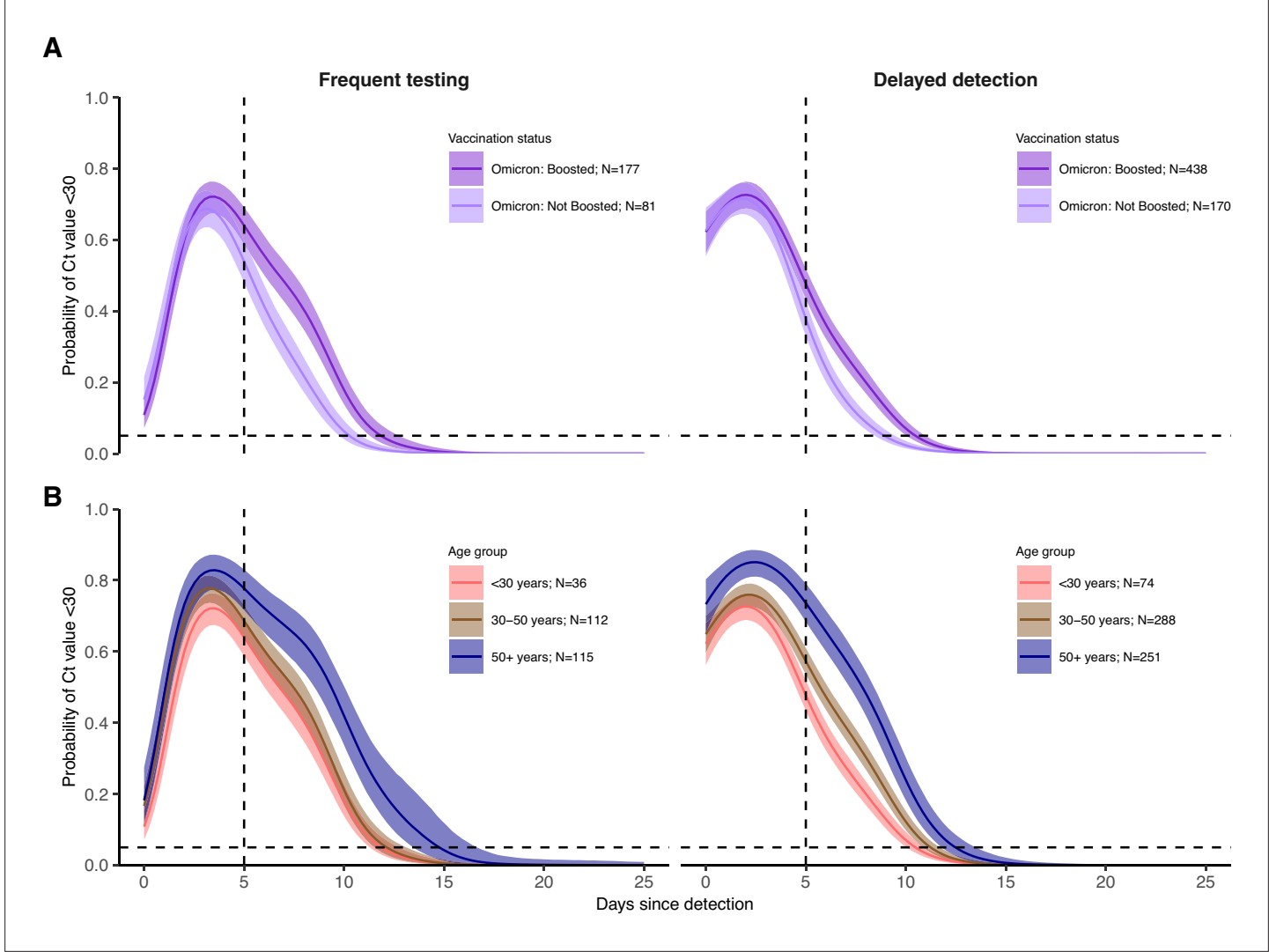

**Figure 2.** Posterior estimates from a generalized linear model predicting probability of Ct value <30 with spline terms for the interaction between days since detection with age group and the interaction between days since detection with vaccination status and variant. Shown are the marginal effects of (**A**) vaccination status and (**B**) age group on the proportion of individuals with Ct <30 on each day post detection after conditioning on being an Omicron BA.1 infection and (**A**) <30 years old (**B**) boosted at the time of infection. Solid colored lines and shaded ribbons show the posterior mean (solid line) and 95% credible intervals (shaded ribbon) of each conditional effect. Dotted horizontal and vertical lines mark 5% probability and day 5 post detection, respectively.

for the frequent testing group and an overall classification accuracy of 85.0% with an AUC of 91.4% (Ct <30 = 71.2%; Ct ≥30/negative = 89.3%) in the delayed detection group. These results indicate that while exposure histories help to explain mean viral RNA kinetics, they provide little assistance in predicting an individual's course of infectiousness over time, due to a high degree of individual-level variation, which may be dominated by stochastic effects or other unmeasured characteristics.

Vaccination provides multiple layers of protection against SARS-CoV-2, leading to reduced rates of infection (*Tai et al., 2022*) and faster clearance of the virus (*Kissler et al., 2021b*). Consistent with these findings, individuals who received two vaccine doses prior to infection with pre-Delta and pre-Omicron variants (N=12) cleared to negative results or high Ct values faster than unvaccinated individuals (N=209) (*Appendix 1—figure 9*). While boosting reduced rates of infection in our cohort,(*Tai et al., 2022*) boosted individuals with Omicron BA.1 infections (N=615) tended to sustain low Ct values for longer durations than individuals who had only undergone an initial vaccine course (N=251), defined as either two doses of an mRNA vaccine or a single dose of the Ad.26.COV2.S adenovirus vector-based vaccine (*Figure 2A*; *Appendix 1—figure 9*). This pattern

was robust to refitting the model after excluding player infections, resulting in a subpopulation more representative of the general population in age and health status (*Appendix 1—figure 10*). We also found similar patterns after subsetting infections by their symptom status (*Appendix 1—figure 11*).

It is important to consider the possible confounding effect of age, as boosted individuals in this cohort were typically older than non-boosted individuals at the time of BA.1 infection (mean age of 37.6 years in the BA-1 infected, boosted group vs. 31.3 years in the BA.1-infected, non-boosted group). The regression model including age group, vaccination status and variant found that older individuals do maintain Ct <30 for longer on average than younger individuals after conditioning on vaccination status (*Figure 2B*). However, the effect of a higher proportion with Ct <30 in boosted individuals relative to non-boosted individuals also remained within each age group, suggesting that both older age and booster status explain some variation in duration of Ct <30 (*Appendix 1—figure 12*). Furthermore, models including age were almost universally better supported in the model comparison analysis and provided improvements in classification accuracy, but in both cases the gains were small (*Appendix 1—table 3* and *Appendix 1—table 4*).

## Pre-Omicron antibody titer explains variation in viral RNA clearance

To assess the mechanisms behind the unexpected slower clearance in boosted Omicron BA.1 infections, we analyzed viral kinetics stratified by antibody titer. In addition to exposure history information, 979 individuals were tested at least once (1,017 measurements total) with the Diasorin Trimeric Assay for antibody titers against the spike protein from the ancestral SARS-CoV-2 (WA1) strain (*Appendix 1—figure 13*). Most titers were obtained from mid-September to mid-October 2021, and thus we consider these titers to represent an individual's post primary vaccination course response rather than post-boost/post-Omicron infection immunity (*Appendix 1—figure 2*). The median time between the most recent vaccine dose and the titer draw was 162 days (interquartile range: 129–180 days) (*Appendix 1—figure 14*).

We hypothesized that these single point-in-time SARS-CoV-2 antibody titer measurements represented a proxy of the strength of the immune response to SARS-CoV-2 and thus would be reflected in the features of viral kinetics over the course of infection. A total of 494 measurements were classified as low antibody titers (≤250 arbitrary units [AU]/ml) and 523 as high titers (>250 AU/ml). This cutoff was chosen as a conservative upper bound for defining risk of Delta infection.

We fitted a logistic regression model for the probability of having Ct <30 on each day since detection, stratified by the interaction of an individual's booster status and their pre-booster antibody titer status, as well as an additional stratification by age group (*Figure 3A*). Boosted individuals with a low antibody titer had the highest and longest duration of Ct <30 over time since detection in both the frequent testing and delayed detection group. In the delayed detection group, individuals with low antibody titers were more likely to have Ct <30 than individuals with high antibody titers regardless of booster status, though boosted individuals with high antibody titers maintained Ct <30 for longer than non-boosted individuals with low titers.

The results were consistent with an age group-level effect also contributing towards differences in the proportion of individuals with Ct <30 over time. Trends were similar within each age group, but we note that at this level of stratification the sample sizes for some subgroups are small and thus there is considerable uncertainty for some combinations of age group, titer group, and vaccination status. We found that younger BA.1-infected individuals had higher antibody titers on average than older BA.1-infected individuals, but that BA.1-infected boosted individuals had consistently lower mean antibody titers than BA.1-infected non-boosted individuals within each age group (*Appendix 1—figure 15*).

To account for potential confounding from waning immunity, in which low titers simply represent a longer time since previous exposure, we restricted the dataset to include only individuals who had their titer measured within 100–200 days of a previous exposure with the aim of comparing antibody titers measured at a similar point in the waning process. We also repeated the analysis after restricting to only infections detected 60–90 days following an antibody titer measurement, here aiming to include only infections for which the measured titer reasonably proxies the titer at the time of infection. These time windows were chosen to improve comparability of immune states while also retaining reasonably large sample sizes. The trend of a higher and longer duration of Ct <30 in Omicron BA.1-infected, boosted, low titer individuals was maintained in both sensitivity analyses (*Appendix 1—figure 16*).

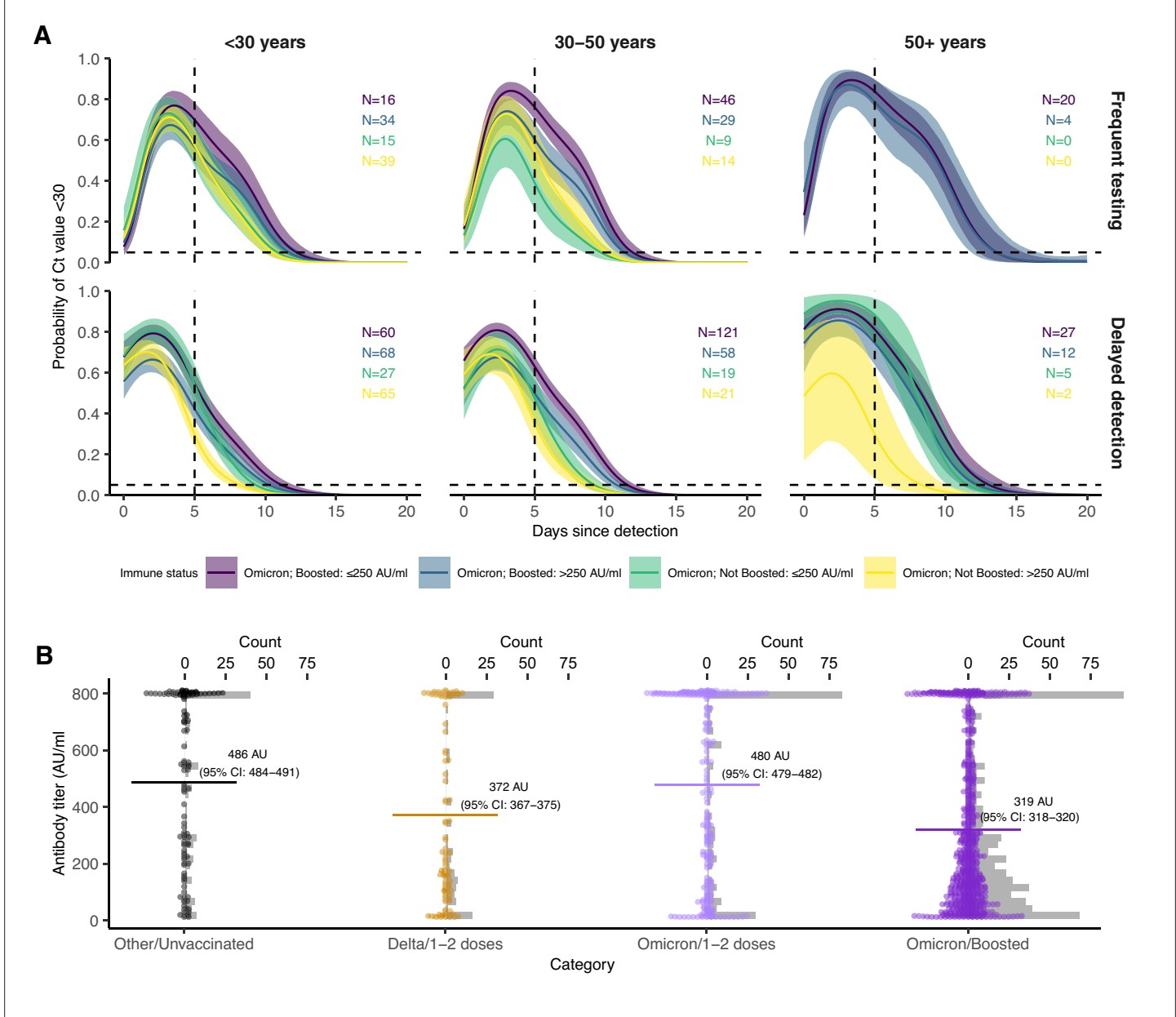

**Figure 3.** Effect of booster status, antibody titer against the founder SARS-CoV-2 strain and age group on the probability of Ct value <30 over days post detection. (**A**) Proportion of Omicron BA.1 infections with Ct value <30 on each day post detection, stratified by age group and the interaction of booster status at the time of infection and antibody titer group. Shown are posterior estimates from a generalized linear model predicting probability of Ct <30 with a spline term on days since detection, conditional on titer/vaccination status category and age group. Solid lines show posterior mean and shaded ribbons show 95% credible intervals of each conditional effect. (**B**) Distribution of measured antibody titers (colored points) stratified by variant and vaccination status of each detected infection, with mean titers (horizontal lines) and bootstrapped 95% confidence intervals (CIs) shown in text. Note that the 95% CIs are very small relative to the range and are thus not plotted. Grey bars are histograms of antibody titer counts in bins of 10 arbitrary units (AU)/ml. Note also that stratification is by infection event and not individual, and that antibody titers were measured at a single point in time rather than near the time of infection. The Diasorin Trimeric Assay values are truncated between 13 and 800 AU/ml.

Based on these findings, we hypothesized that boosted individuals who nevertheless were infected with Omicron BA.1 may have had relatively poor BA.1-specific immune responses to prior SARS-CoV-2 exposures, leading to longer infection durations. This is demonstrated by stratifying antibody titers by variant and vaccination status at the time of infection (*Figure 3B*). Antibody titers were lower among fully vaccinated individuals who were subsequently infected with Delta than individuals who had been infected with a pre-Delta variant. This suggests that individuals with a high antibody titer at

around the time of Delta circulation were less likely to be infected with Delta. In contrast, we found that mean antibody titers among Omicron BA.1-infected, fully vaccinated individuals were similar to individuals in the pre-Delta, unvaccinated group, suggesting that higher titer individuals were not substantially less likely to be infected than lower titer individuals. Finally, we found that antibody titers were lowest among Omicron BA.1 infected boosted individuals, suggesting that individuals with a high titer measurement prior to being boosted were less likely to have Omicron BA.1 infections.

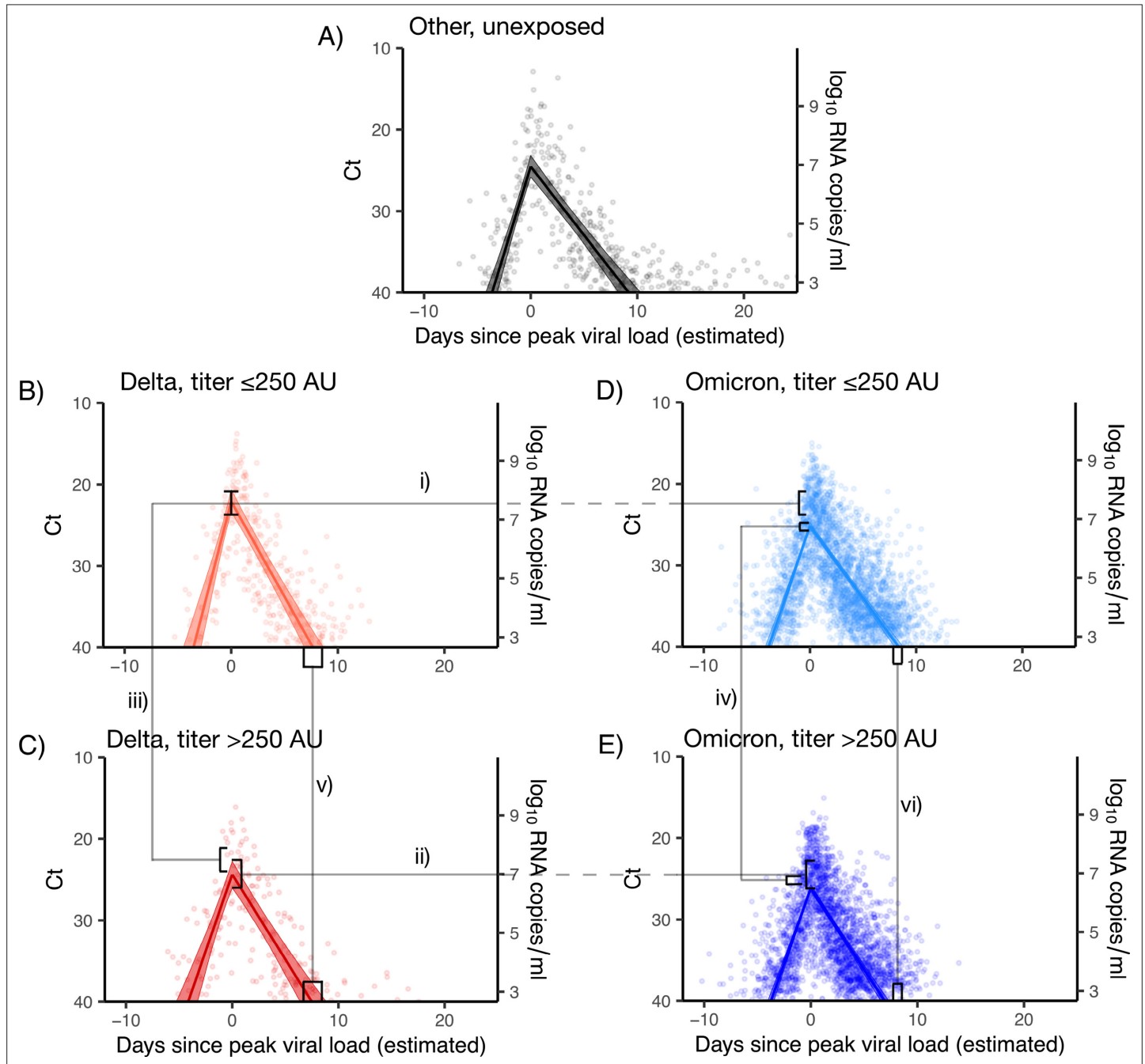

**Figure 4.** Estimated viral trajectories by variant and titer. Points depict measured Ct values, lines depict the estimated population mean viral trajectories, and shaded regions depict the 95% credible intervals for the estimated population viral trajectories. (**A**) Non-Delta and non-Omicron infections in individuals who were previously unexposed (no prior record of vaccination or infection), (**B**) Delta infections with titer ≤250, (**C**) Delta infections with titer >250, (**D**) Omicron BA.1 infections with titer ≤250, (**E**) Omicron BA.1 infections with titer >250. Peak viral loads were higher for Delta infections than for Omicron BA.1 infections when stratifying by titer (**i and ii**), and titers ≤250 were associated with higher viral loads when stratifying by variant (iii and iv). Low titers were also associated with longer clearance times (**v and vi**).

## The effect of immune status and variant on viral proliferation, peak viral RNA titers, and clearance

We next adapted a framework to estimate the impact of antibody titer, vaccination status, and variant on peak viral RNA concentrations, proliferation phase duration, and clearance duration (*Figure 4*; *Kissler et al., 2021a*; *Kissler et al., 2021b*). According to the viral kinetic model, and among Omicron BA.1 infections, boosted individuals had a longer estimated viral clearance time than non-boosted individuals (8.4 days (95% CrI: 8.0–8.7) *vs.* 6.2 days (95% CrI: 5.8–6.6), respectively), in line with the results from the logistic regression model. Viral proliferation times and peak viral RNA were similar among boosted and non-boosted individuals with Omicron BA.1 infections (*Appendix 1—table 5*). When stratifying by post-initial vaccination antibody titer, Delta infections featured a consistently higher peak viral RNA than Omicron BA.1 infections. Among Omicron BA.1 infections, high antibody titers were associated with faster viral clearance times and lower peak viral RNA. Proliferation times were similar across variants and titers (*Appendix 1—table 6*).

We fitted the viral RNA kinetic model to Omicron BA.1 infections after stratifying individuals based on their symptom status as well as vaccination status or antibody titer group. We found the same pattern of longer clearance times for boosted individuals relative to fully vaccinated individuals, with symptomatic boosted individuals demonstrating longer clearance times than asymptomatic boosted individuals (*Appendix 1—table 7*). Among those with low antibody titer, presence of symptoms was associated with higher peak viral RNA and longer clearance times, while for those with high antibody titer, peak viral RNA and clearance times were similar between symptom statuses (*Appendix 1—table 8*).

Finally, we fitted the models allowing for BA.1 viral RNA kinetics to vary by age in addition to vaccination status or antibody titer. Consistent with the logistic regression results, older individuals demonstrated longer average clearance times than younger individuals across vaccination and antibody titer groups. When stratifying by immune status, individuals aged 50+years took between roughly 1–2 days longer to clear than individuals aged 30–50, and 2–3 days longer to clear than individuals under 30 (*Appendix 1—table 9* and *Appendix 1—table 10*). However, we still found a consistent effect of antibody titer and booster status on clearance time despite this additional age effect. Individuals with low antibody titer and boosted individuals took 1–2 days longer to clear than individuals with high antibody titer and non-boosted individuals in both the <30 and 30–50 year age groups (*Appendix 1—table 9* and *Appendix 1—table 10*).

## Discussion

We found that individuals infected with SARS-CoV-2 often had Ct values <30 beyond the five-day isolation period following SARS-CoV-2 infection currently recommended by the CDC (*Centers for Disease Control and Prevention, 2022*). This finding is in line with other studies measuring Ct values from upper respiratory tract samples, the duration of antigen test positivity, and the duration of infectious viral load or culturable virus (*Boucau et al., 2022a*; *Earnest et al., 2022*; *Ke et al., 2022*; *Landon et al., 2022*; *Lefferts et al., 2022*). While we do not have data on infectiousness by day to clarify the exact link between Ct and infectiousness, nearly half of the individuals in this cohort had potentially infectious viral loads (Ct <30) five days after their initial detection, even in those detected later in their infection course (*Singanayagam et al., 2020*). By day 10, the number of individuals with Ct <30 was substantially reduced but still high. The duration of positivity was highly variable across individuals, and low Ct values consistent with potential infectiousness were sometimes maintained for up to two weeks. These observations suggest the use of test-based, rather than time-based, protocols for defining the duration of isolation to limit the spread of SARS-CoV-2.

Rebounds with recurrence of symptoms and positive rapid antigen tests after a period of negative test results have been increasingly reported in individuals treated with SARS-CoV-2 antiviral drugs (*Boucau et al., 2022b*; *Charness et al., 2022*), but estimates for the frequency of viral rebounds in the absence of antiviral treatment have been lacking. Among infected boosted individuals in this cohort, who were predominantly infected with Omicron BA.1, we detected seven rebounds in viral trajectory, stringently defined as any Ct value trajectory with at least three consecutive days of negative tests or tests with Ct ≥30 after the initial peak followed by 3threeor more consecutive days with Ct <30. However, more rebounds were detected when using less stringent Ct value-based definitions and

were more frequent in Omicron BA.1-infected or boosted individuals, occurring in ~6% of infections in contrast to ~1% of infections in the pre-booster pre-Omicron phase of the pandemic. It was not routine for testing to continue following suspected clearance in this cohort, and thus these results may represent a lower bound on the incidence of rebound infections. The frequency of viral trajectory rebounds depends on the definition of 'rebound', highlighting the need for standardized definitions to enable comparisons across studies. We did not measure the recurrence of culturable virus during these resurgent low Ct periods, and thus further work is needed to understand if viral RNA rebounds are a reliable proxy for infectivity. Moreover, we did not have sufficient information to define rebounds with respect to clearance and recurrence of symptoms, though the experience of the occupational health team is that rebounds of a clinical nature have been extremely rare, with only one documented case (*Mack et al., 2021*). Overall, these findings suggest that symptom monitoring after clearing isolation may be warranted, and a return to isolation may be necessary for individuals with rebound infections (*Charness et al., 2022*).

Boosted individuals in this cohort were less likely to be infected with Omicron BA.1, (*Tai et al., 2022*) and those who had a breakthrough infection tended to have a low antibody titer measurement to the WA1 spike protein after their initial vaccine course. In this context, test positivity following Omicron BA.1 infection lasted longer for boosted individuals than for non-boosted individuals, regardless of symptom status. This observation was further supported by a viral kinetic model that found longer clearance times for Omicron BA.1 infections in boosted relative to non-boosted individuals. Moreover, high antibody titers to the WA1 spike protein were associated with lower peak viral RNA concentrations and faster clearance times for both Delta and Omicron BA.1 infections. Together, these results suggest that the low antibody titers in infected boosted individuals conferred increased risk for infection as well as slower control and clearance of infection.

The effect of age on viral kinetics complicates the interpretation of these findings. Prior to the detection of Omicron BA.1, older individuals have been found to take longer to clear infection on average than younger individuals (*Caputo et al., 2021*; *Cevik et al., 2021*; *Jones et al., 2021*; *Long et al., 2021*; *Néant et al., 2021*; *Singanayagam et al., 2022*). However, these findings are not unequivocal, as a previous systematic review found the effect of age on viral kinetics was diminished after accounting for disease severity (*Chen et al., 2021*). Our data support an effect of age on viral clearance times, with longer times from peak to clearance in individuals >50 years compared to those <30 years regardless of variant and immune state. In this cohort, older individuals were more likely to be boosted prior to becoming infected with BA.1 than younger individuals, and thus the finding of delayed clearance in BA.1-infected, boosted individuals can be partially attributed to delayed clearance in older individuals. However, we found consistent delayed clearance in boosted relative to non-boosted individuals within each age group, notably in the <30 years group. Furthermore, the pattern of lower antibody titers to WA1 spike in BA.1-infected, boosted individuals relative to BA.1-infected, non-boosted individuals was also consistent within each age group, suggesting that low WA1 spike titers correlate with increased infection risk and slower clearance in addition to any age-specific effects.

An important limitation of this study is that the cohort is not representative of the general population, as it is predominately male, young, and includes professional athletes. However, our key findings were preserved in analyses after excluding the players. We did not test for the presence of infectious virus, and our findings are based on Ct values obtained from combined nasal and oropharyngeal swabs.(*Ke et al., 2022*) While low Ct values have been associated with potential infectiousness and antigen test positivity (*Bullard et al., 2020*; *Jaafar et al., 2021*; *Jefferson et al., 2020*; *Singanayagam et al., 2020*), this is an imperfect proxy. It is possible that some infections were undetected, and thus the reported number of prior infections should be interpreted as a lower bound for each member of the cohort. SARS-CoV-2 antibody titers were only measured from mid-September to mid-October 2021 and were taken at varying time points after initial vaccination course (between 0 and 290 days), so we could not assess the relationship between antibody waning and viral kinetics. Antibody titers were measured against the spike protein of the WA1 lineage, which correlate poorly with protection against the antigenically distinct Omicron lineages; thus, it is unclear how these data are associated specifically to Omicron-immunity, beyond representing a proxy for overall immune response.

Variants and immune statuses interact, sometimes in unexpected ways, to produce viral kinetics that differ in duration and intensity. Collecting longitudinal viral load data in more diverse cohorts will

help to ensure that isolation and quarantine policies are based on the best available evidence and will help to properly contextualize results from ongoing drug and vaccine trials. Similarly, our findings suggest that SARS-CoV-2 control measures may be better informed by measurements of immune status than proxies such as number or timing of receipt of vaccine doses or of infections. Testing this hypothesis will require widespread collection and analysis of serological, infection, and vaccination data in diverse cohorts and broader availability of quantitative antibody tests designed for the spike protein of Omicron lineages.

## Acknowledgements

Supported in part by CDC contract #200-2016-91779, a sponsored research agreement to Yale University from the National Basketball Association contract #21–003529, and the National Basketball Players Association.

## Additional information

### Competing interests

Stephen M Kissler: SMK has a consulting agreement with the NBA. Joseph R Fauver, Nathan D Grubaugh: has a consulting agreement for Tempus and receives financial support from Tempus to develop SARS-CoV-2 diagnostic tests. Christina Mack, Caroline G Tai, Radhika M Samant, Sarah Connolly: is an employee of IQVIA, Real World Solutions. Deverick J Anderson: is co-owner of Infection Control Education for Major Sports. Gaurav Khullar, Matthew MacKay, Miral Patel, Shannan Kelly, April Manhertz, Isaac Eiter, Daisy Salgado, Tim Baker, Ben Howard, Joel T Dudley, Christopher E Mason: is an employee of Tempus Labs. John DiFiori: is an employee of the NBA. Yonatan H Grad: has a consulting agreement with the NBA. The other authors declare that no competing interests exist.

### Funding

| Funder | Grant reference number | Author |
| --- | --- | --- |
| Centers for Disease Control and Prevention | 200-2016-91779 | Yonatan H Grad |
| National Basketball Association | 21-003529 | Nathan D Grubaugh |

The funders had no role in study design, data collection and interpretation, or the decision to submit the work for publication.

### Author contributions

James A Hay, Stephen M Kissler, Conceptualization, Data curation, Software, Formal analysis, Validation, Investigation, Visualization, Methodology, Writing – original draft, Writing – review and editing; Joseph R Fauver, Data curation, Formal analysis, Validation, Investigation, Methodology, Writing – review and editing; Christina Mack, Conceptualization, Data curation, Formal analysis, Investigation, Methodology, Writing – original draft, Project administration, Writing – review and editing; Caroline G Tai, Data curation, Formal analysis, Investigation, Methodology, Project administration, Writing – review and editing; Radhika M Samant, Sarah Connolly, Gaurav Khullar, Matthew MacKay, Miral Patel, Shannan Kelly, April Manhertz, Isaac Eiter, Daisy Salgado, Tim Baker, Ben Howard, Joel T Dudley, Christopher E Mason, Data curation, Writing – review and editing; Deverick J Anderson, John DiFiori, Investigation, Writing – review and editing; Manoj Nair, Yaoxing Huang, Data curation, Formal analysis, Investigation, Writing – review and editing; David D Ho, Formal analysis, Investigation, Writing – review and editing; Nathan D Grubaugh, Conceptualization, Resources, Data curation, Formal analysis, Supervision, Funding acquisition, Validation, Investigation, Visualization, Methodology, Project administration, Writing – review and editing; Yonatan H Grad, Conceptualization, Resources, Formal analysis, Supervision, Funding acquisition, Validation, Investigation, Visualization, Methodology, Writing – original draft, Project administration, Writing – review and editing

## Author ORCIDs

James A Hay ![ORCID] http://orcid.org/0000-0002-1998-1844
Stephen M Kissler ![ORCID] http://orcid.org/0000-0003-3062-7800
Manoj Nair ![ORCID] http://orcid.org/0000-0002-5994-3957
Yonatan H Grad ![ORCID] http://orcid.org/0000-0001-5646-1314

## Decision letter and Author response

Decision letter https://doi.org/10.7554/eLife.81849.sa1
Author response https://doi.org/10.7554/eLife.81849.sa2

## Additional files

### Supplementary files
• MDAR checklist

### Data availability
All code and data required to reproduce the analyses are available at https://github.com/gradlab/SC2-kinetics-immune-history, (copy archived at swh:1:rev:4cd9f83213c178d148ae59a245f93beec3ace825).

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

## Appendix 1

### Logistic regression models

#### Model fitting

We fitted Bayesian logistic regression models for the probability of an individual having Ct value <30 on each day post detection using the *brms* package version 2.14.4. Models were run on the Harvard FAS Research Computing cluster using R version 4.0.2. For each model, we ran four chains for 2000 iterations each. Weakly informative priors (normal distributions with means of 0 and standard deviations of 10) were used for all model parameters. We assessed convergence based on all estimated parameters having a Gelman R-hat statistic less than 1.1.

### Viral kinetics model

#### Statistical analysis

Following previously described methods, (*Kissler et al., 2021a*; *Kissler et al., 2021b*) we used a Bayesian hierarchical model to estimate the proliferation duration, clearance duration, and peak viral concentration for acute SARS-CoV-2 infections, stratified by variant (Omicron, Delta, Other), immune status (vaccination history, including unexposed, 1–2 doses, or boosted; and antibody titer, including unexposed, titer ≤250 AU, and titer >250 AU) and age (groups of <30, 30–50, and >50 years old). The model describes the $\log_{10}$ viral concentration during an acute infection using a continuous piecewise-linear curve with control points that specify the time of acute infection onset, the time and magnitude of peak viral concentration, and the time of acute infection clearance. The assumption of piecewise linearity is equivalent to assuming exponential viral growth during the proliferation period followed by exponential viral decay during the clearance period. The control points were inferred using the Hamiltonian Monte Carlo algorithm as implemented in Stan (version 2.24) (*Carpenter, 2017*). We used priors informed by a previous analyses (*Kissler et al., 2021a*; *Kissler et al., 2021b*). Data and code are available online.

#### Model fitting

To restrict to a set of well-observed acute infections for model fitting, we first removed any sequences of three or more consecutive negative tests (Ct = 40) from each acute infection to avoid overfitting to these trivial values. We kept only acute infections with at least one Ct value <32 and at least three Ct values <40 (the limit of detection).

We constructed a piecewise-linear regression model to estimate the peak Ct value, the time from infection onset to peak (*i.e.* the duration of the proliferation stage), and the time from peak to infection resolution (*i.e.* the duration of the clearance stage). This is represented by the equation

$$\text{E}[Ct(t)] = \begin{cases} \text{l.o.d} - \frac{\delta}{t_p - t_o}(t - t_o) & t \leq t_p \\ \text{l.o.d} - \delta + \frac{\delta}{t_p - t_o}(t - t_p) & t > t_p \end{cases}$$

Here, E[*Ct(t)*] represents the expected value of the Ct at time *t*, "l.o.d" represents the RT-qPCR limit of detection, $\delta$ is the absolute difference in Ct between the limit of detection and the peak (lowest) Ct, and $t_o$, $t_p$, and $t_r$ are the onset, peak, and recovery times, respectively.

Before fitting, we re-parametrized the model using the following definitions:

- $\Delta Ct(t)$=l.o.d. – *Ct(t)* is the difference between the limit of detection and the observed Ct value at time *t*.
- $\omega_p = t_p\ t_o$ is the duration of the proliferation stage.
- $\omega_r = t_r\ t_p$ is the duration of the clearance stage.

We next characterized the likelihood of observing a given $\Delta Ct(t)$ using the following mixture model:

$$L(\Delta Ct(t) = x|\delta, t_p, \omega_p, \omega_r) = (1 - \lambda)[f_N(x|E[\Delta Ct(t)], \sigma(t)) + I_{lod}F_N(0|E[\Delta Ct(t)], \sigma(t))] + \lambda f_{Exp}(x|k)$$

The left-hand side of the equation denotes the likelihood (*L*) that the observed viral load, as measured by Ct deviation from the limit of detection (Δ*Ct(t)*), is equal to some quantity *x* given the model parameters *δ* (peak viral load), $t_p$ (time of peak viral load), $\omega_p$ (proliferation time), and $\omega_r$ (clearance time). This likelihood is equal to the sum of two main components: the likelihood that the observed value was generated by the modeled viral kinetic process, denoted by the bracketed term preceded by a (1-*λ*); and the likelihood that the observed value was a false negative, denoted by the term preceded by a *λ*. In the bracketed term representing the modeled viral kinetic process, $f_N(x \mid E[\Delta Ct(t)], \sigma(t))$ represents the Normal PDF evaluated at *x* with mean $E[\Delta Ct(t)]$ (generated by the model equations above) and observation noise $\sigma(t)$. $F_N(0 \mid E[\Delta Ct(t)], \sigma(t))$ is the Normal CDF evaluated at 0 with the same mean and standard deviation. This represents the scenario where the true viral load goes below the limit of detection, so that the observation sits at the limit of detection. $I_{lod}$ is an indicator function that is 1 if Δ*Ct(t)*=0 and 0 otherwise; this way, the $F_N$ term acts as a point mass concentrated at Δ*Ct(t)*=0. Last, $f_{Exp}(x \mid \kappa)$ is the Exponential PDF evaluated at *x* with rate *κ*. We set *κ*=log(10) so that 90% of the mass of the distribution sat below 1 Ct unit and 99% of the distribution sat below 2 Ct units, ensuring that the distribution captures values distributed at or near the limit of detection. We did not estimate values for *λ* or the exponential rate because they were not of interest in this study; we simply needed to include them to account for some small probability mass that persisted near the limit of detection to allow for the possibility of false negatives. A schematic of the likelihood function is depicted in ***Appendix 1—figure 17***

We used a hierarchical structure with a non-centered parameterization to describe the distributions of $\omega_p$, $\omega_r$, and *δ* for each person:

$$\omega_p[i] = Exp[\mu_{\omega p} + \zeta^i_{wp} + \sigma_{wp} N^i_{wp}] \, \omega^*_p$$
$$\omega_r[i] = Exp[\mu_{\omega r} + \zeta^i_{wr} + \sigma_{wr} N^i_{wr}] \, \omega^*_r$$
$$\delta[i] = Exp[\mu_\delta + \zeta^i_\delta + \sigma_\delta N^i_\delta] \, \delta^*$$

Here, $\omega^*_p$, $\omega^*_r$, and $\delta^*$ are user-defined estimated values for the means of $\omega_p$, $\omega_r$, and *δ*, so that the exponential terms represent an adjustment factor relative to that midpoint defined by *μ* (a shared adjustment factor for the entire population), $\zeta^i$ (an adjustment factor shared among individuals of a given variant/immune category), and σ (a shared standard deviation for the entire population). The $N^i$ terms represent individual-level random effects. The prior distributions for the *μ*, *ζ*, and σ terms were all Normal(0, 0.25) (with σ truncated to have support on the positive reals). These prior distributions define LogNormal adjustment factors that have ~99% of their probability mass between 0.5 and 2, so that the prior distributions for $\omega_p$, $\omega_r$, and *δ* cover roughly half to twice their prior estimated midpoint values.

We used a Hamiltonian Monte Carlo fitting procedure implemented in Stan (version 2.24) and R (version 3.6.2) to estimate the parameters. We ran four MCMC chains for 2,000 iterations each with a target average proposal acceptance probability of 0.8. The first half of each chain was discarded as the warm-up. The Gelman R-hat statistic was less than 1.1 for all parameters. This indicated good overall mixing of the chains. There were no divergent iterations, indicating good exploration of the parameter space.

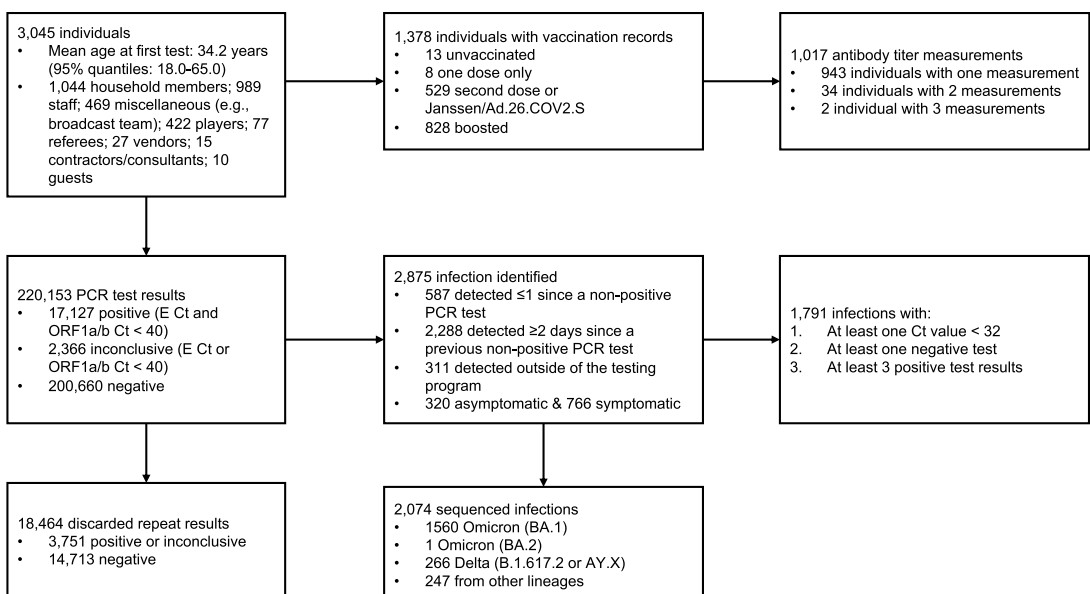

**Appendix 1—figure 1.** Summary of cohort. Top row describes cohort demographics and data on immune histories. Middle row describes infection data. Bottom row provides additional information on the infection data.

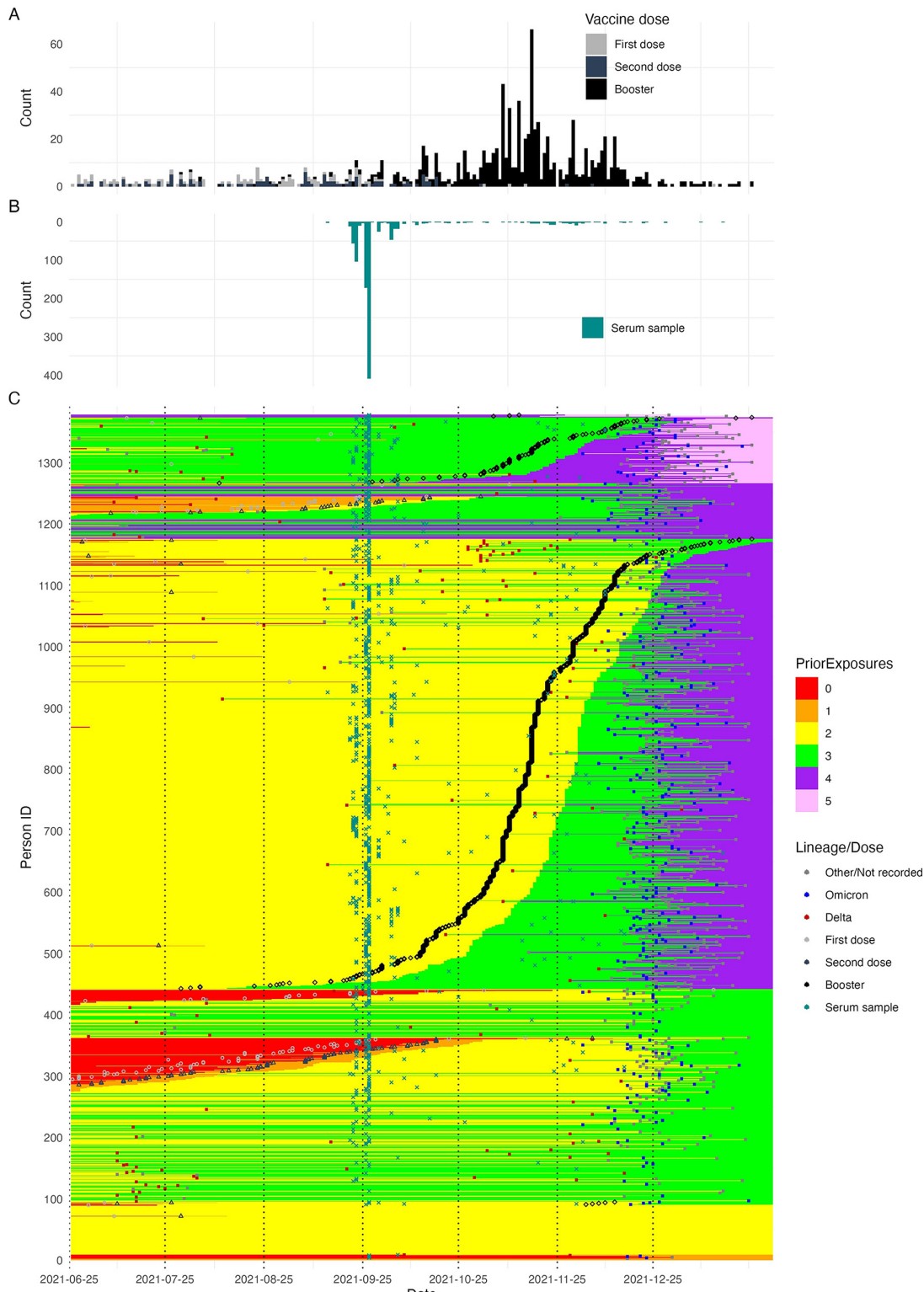

**Appendix 1—figure 2.** Summary of all infection and vaccination events over time, in addition to serum sample collection dates, included in the dataset. (**A**) Histogram showing the distribution of vaccination dates (note that most first doses were administered prior to 2021-06-25), showing when in time the majority of individuals were vaccinated. (**B**) Histogram showing the timing of serum sample collections in the cohort. (**C**) Heatmap showing the entire time course of infection and vaccination histories for each individual in the cohort. Each row represents one individual and columns represent date. Each cell is shaded by the number of prior exposures at that date, showing

*Appendix 1—figure 2 continued on next page*

*Appendix 1—figure 2 continued*

how each individual's cumulative exposure history increases over time. Points show the timing of each detected infection, recorded vaccination, and serum sample collection date. Points are colored by the variant or vaccination number of that exposure.

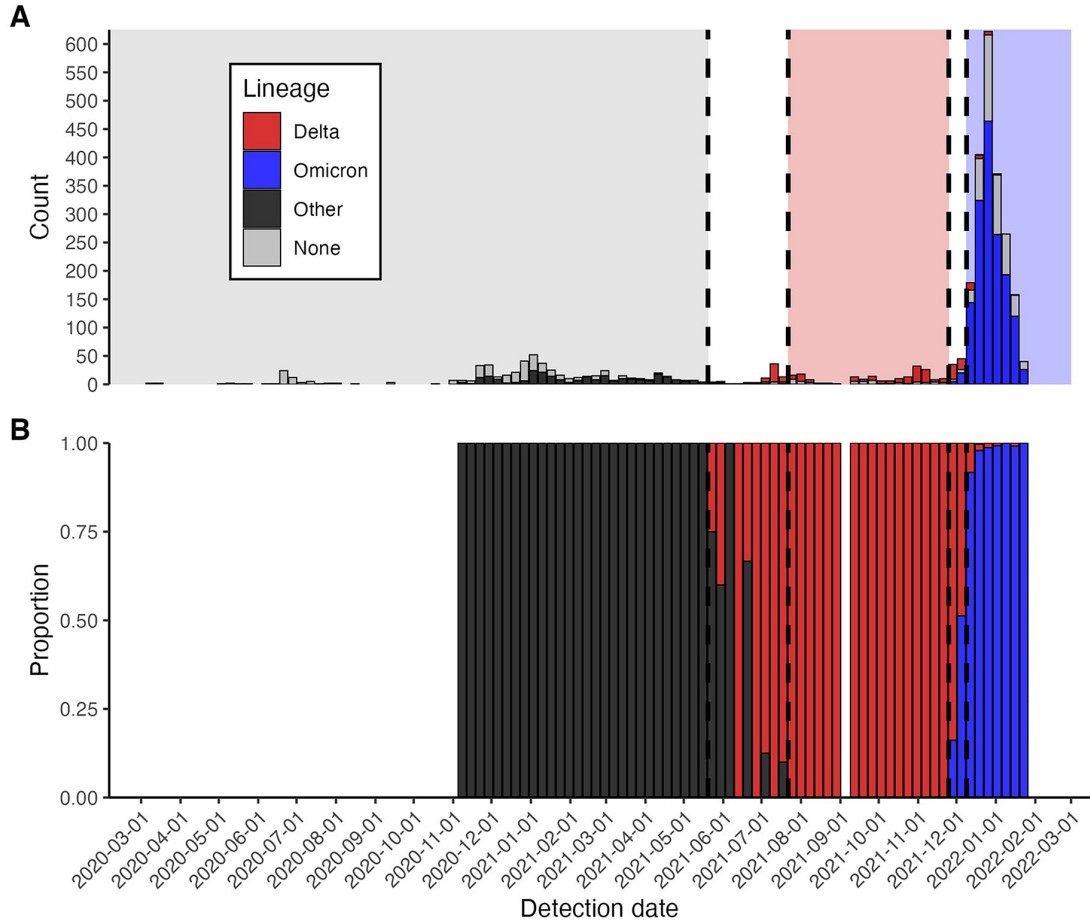

**Appendix 1—figure 3.** Frequency and proportion of SARS-CoV-2 variant detections. (**A**) Frequency of sequenced and unsequenced detected infections over time by week. Vertical dashed lines and shaded backgrounds demarcate periods of variant dominance.

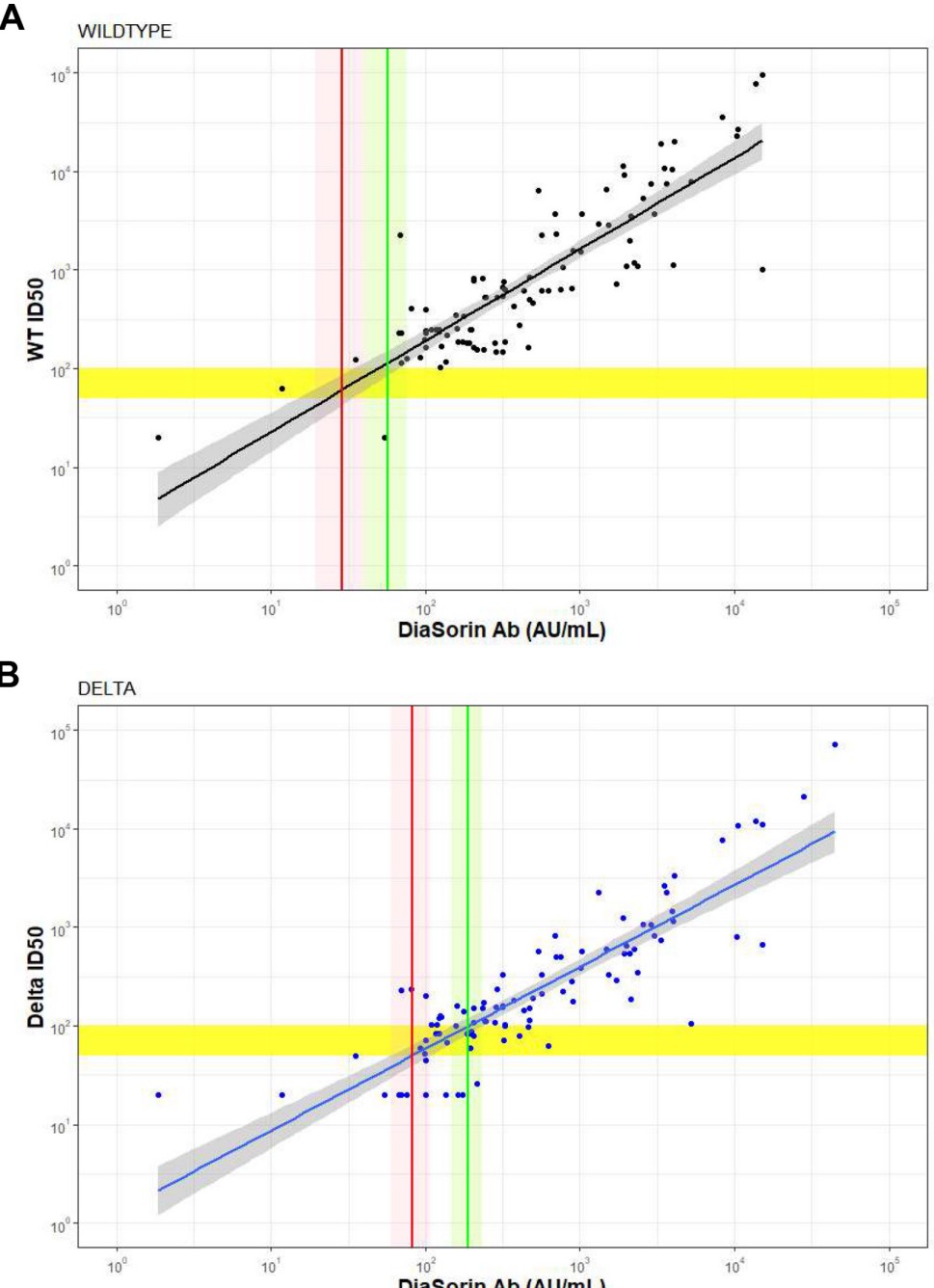

**Appendix 1—figure 4.** Correlation between authentic virus neutralization assay (ID50) and the Diasorin antibody titer against (**A**) wildtype and (**B**) Delta. Horizontal yellow bar shows an ID50 titer of 50 and 100 respectively, Diagonal lines and shaded regions show mean and 95% confidence intervals (CI) for a linear regression between the Diason antibody titer and ID50 titer. Vertical line and shaded regions show point estimate and 95% CI for the Diasorin antibody titer corresponding to an ID50 titer of 50 (red) and 100 (green).

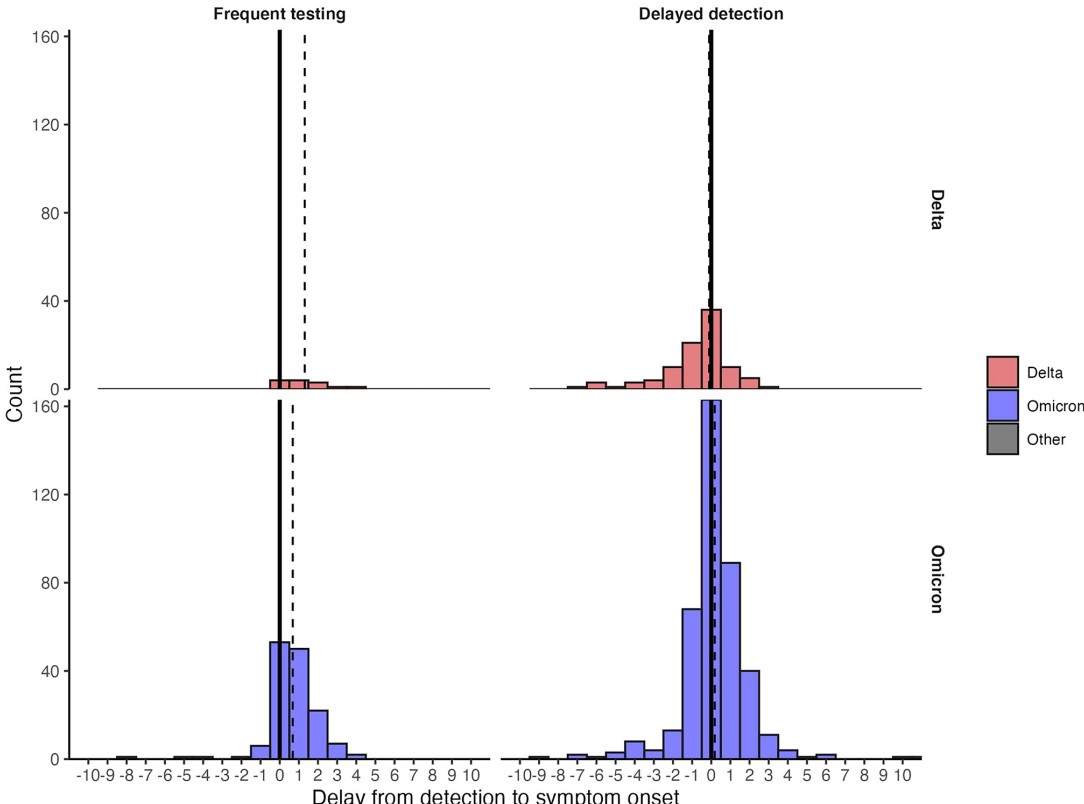

**Appendix 1—figure 5.** Distribution of delays from detection to symptom onset among individuals with known symptom status. Dashed lines mark the median delay between detection and symptom onset. Solid lines mark the day of detection (0).

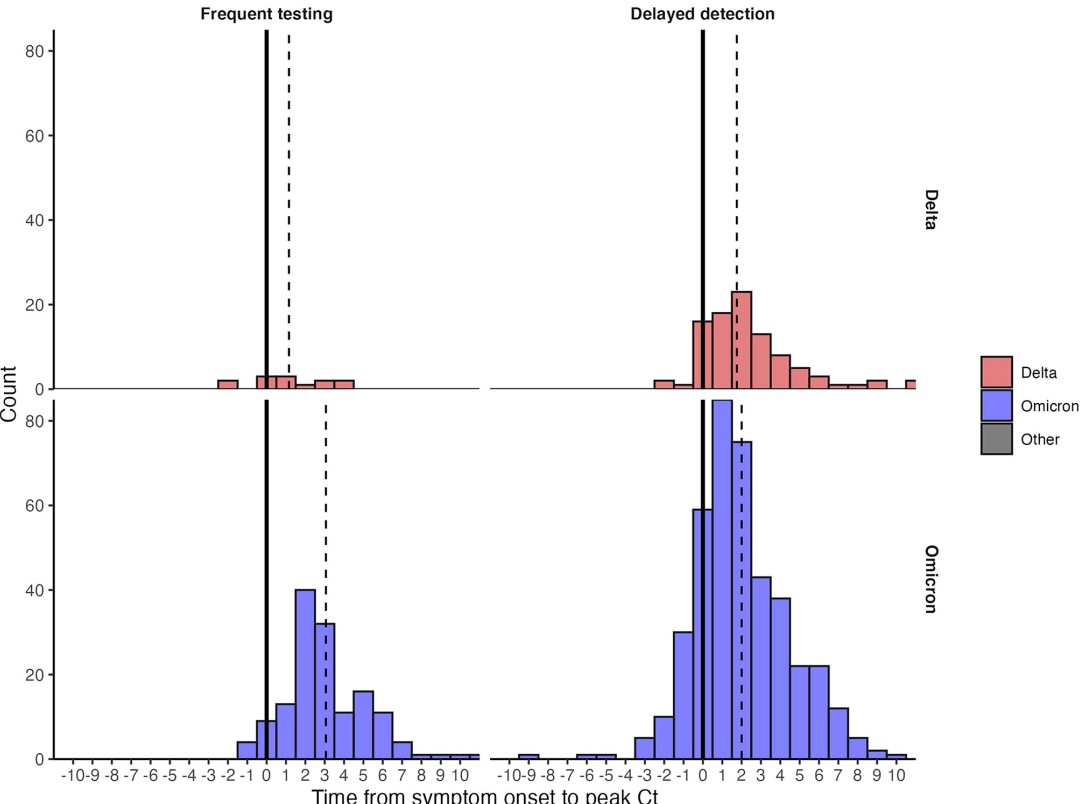

**Appendix 1—figure 6.** Distribution of delays from symptom onset to peak Ct values among individuals with known symptom status. Dashed lines mark the median delay between detection and symptom onset. Solid lines mark the day of symptom onset (0).

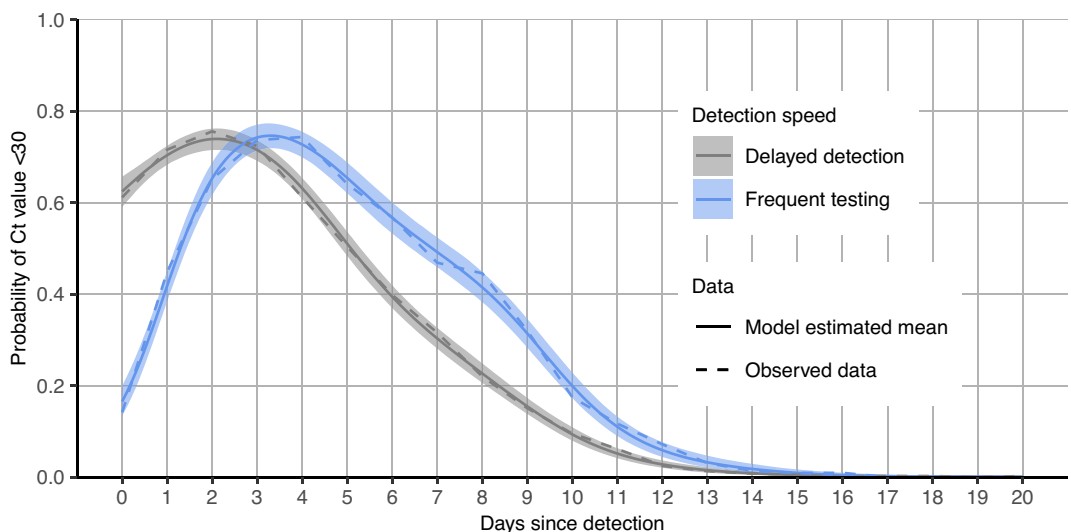

**Appendix 1—figure 7.** Proportion of infections with Ct <30 on each day post detection stratified by detection speed. Solid colored lines and shaded ribbons show posterior means and 95% credible intervals from a generalized linear model predicting probability of Ct <30 as a function of days since detection. Dashed lines show proportion with Ct <30 from the observed data.

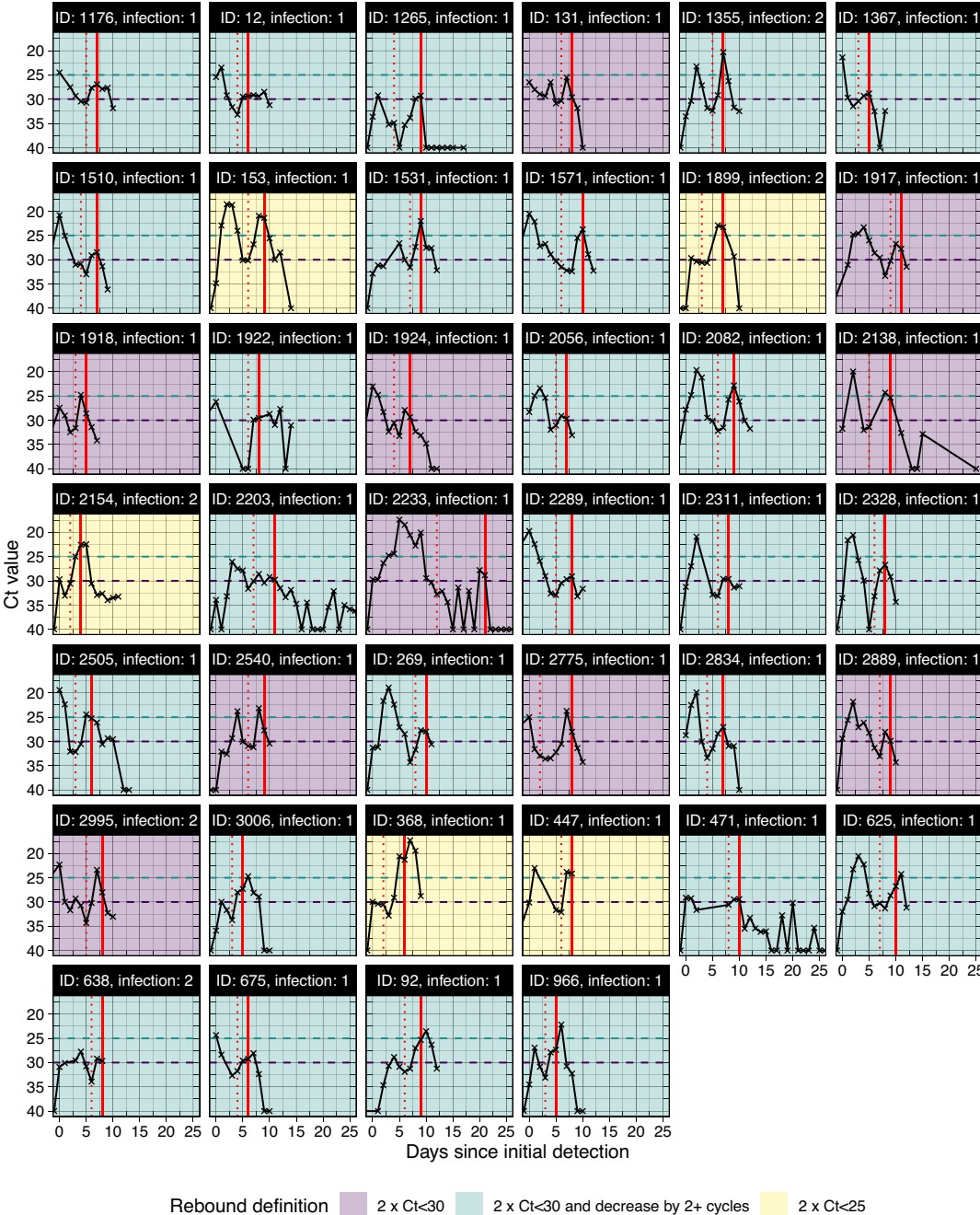

**Appendix 1—figure 8.** All viral trajectories classified as rebound shown in *Figure 1B*. Subplots are colored by the most stringent definition for rebound. To be included here, individuals must have 2+consecutive days of Ct ≥30 after an initial Ct <30. The vertical red dotted line marks this initial clearance time. Trajectories are then classified as rebounds following either two consecutive tests with Ct <30 (purple), two consecutive tests with Ct <30 but with at least a 2 Ct decrease (green), or two consecutive tests with Ct <25 (yellow). The vertical red line marks the timing of rebound detection. The horizontal dashed lines show the different Ct value thresholds for rebound classification. Panels are labeled by arbitrary person ID and infection number. (**B**) Proportion of sequenced infections attributable to Delta, Omicron BA.1 or other variants.

**Appendix 1—table 1.** Number of identified rebounds stratified by variant, either confirmed through sequencing or assumed based on detection date.
Rebounds are defined here as any trajectory with an initial Ct <30, followed by a sequence of two

or more consecutive negative tests or tests with Ct ≥30, and subsequently followed by two or more consecutive tests with Ct <30.

| Variant | Rebounds | Total infections | Percentage rebounded |
|---|---|---|---|
| Omicron | 36 | 877 | 4.10% |
| Delta | 1 | 178 | 0.562% |
| Other | 3 | 279 | 1.08% |

**Appendix 1—table 2.** Number of identified rebounds stratified by vaccination status.
Rebounds are defined here as any trajectory with an initial Ct <30, followed by a sequence of two or more consecutive negative tests or tests with Ct ≥30, and subsequently followed by two or more consecutive tests with Ct <30.

| Vaccination status | Rebounds | Total infections | Percentage rebounded |
|---|---|---|---|
| Boosted | 32 | 494 | 6.48% |
| No record | 5 | 398 | 1.26% |
| Second dose | 3 | 323 | 0.929% |
| Unvaccinated | 2 | 159 | 1.26% |

**Appendix 1—table 3.** Comparison of linear logistic regression models predicting probability of Ct <30 on each day since detection among individuals in the frequent testing group.
Models are ranked based on their expected log pointwise predictive density (ELPD), where a lower ELPD implies better predictive accuracy. Model weight refers to the weight of each model in a Bayesian Model Averaging analysis, where a higher value implies a greater contribution to model prediction when combining multiple models. AUC = area under the curve.

| Model | ELPD difference | SE difference | Weight | AUC | Classification accuracy | Accuracy (≥30) | Accuracy (<30) |
|---|---|---|---|---|---|---|---|
| Cumulative number of exposures, variant and age | 0.000 | 0.000 | 0.157 | 0.897 | 0.828 | 0.890 | 0.647 |
| Days since previous exposure, variant and age | −1.813 | 10.011 | 0.299 | 0.896 | 0.823 | 0.878 | 0.664 |
| Vaccination status, variant and age | −7.440 | 9.828 | 0.089 | 0.896 | 0.824 | 0.881 | 0.656 |
| Cumulative number of exposures and age | −27.935 | 8.823 | 0.104 | 0.894 | 0.823 | 0.887 | 0.637 |
| Cumulative number of exposures and variant | −31.536 | 9.428 | 0.000 | 0.894 | 0.824 | 0.891 | 0.630 |
| Vaccination status and age | −33.933 | 12.389 | 0.000 | 0.893 | 0.822 | 0.882 | 0.646 |
| Vaccination status and variant | −35.274 | 13.693 | 0.156 | 0.893 | 0.824 | 0.886 | 0.645 |
| Days since previous exposure and variant | −39.750 | 14.206 | 0.000 | 0.891 | 0.822 | 0.883 | 0.644 |
| Days since previous exposure and age | −40.929 | 12.503 | 0.000 | 0.892 | 0.820 | 0.877 | 0.654 |
| Variant and age | −52.437 | 11.681 | 0.000 | 0.890 | 0.817 | 0.884 | 0.624 |
| Cumulative number of exposures | −53.661 | 12.344 | 0.143 | 0.892 | 0.822 | 0.895 | 0.610 |
| Vaccination status | −58.503 | 15.162 | 0.000 | 0.890 | 0.822 | 0.888 | 0.632 |
| Age | −60.160 | 12.688 | 0.000 | 0.890 | 0.818 | 0.882 | 0.632 |
| Days since previous exposure | −69.842 | 15.547 | 0.045 | 0.888 | 0.817 | 0.884 | 0.624 |
| Variant | −87.427 | 15.554 | 0.007 | 0.887 | 0.818 | 0.889 | 0.611 |
| Baseline | −93.087 | 15.843 | 0.000 | 0.886 | 0.817 | 0.890 | 0.606 |

**Appendix 1—table 4.** Comparison of linear logistic regression models predicting probability of Ct <30 on each day since detection among individuals in the delayed testing group.

Models are ranked based on their expected log pointwise predictive density (ELPD), where a lower ELPD implies better predictive accuracy. Model weight refers to the weight of each model in a Bayesian Model Averaging analysis, where a higher value implies a greater contribution to model prediction when combining multiple models. AUC = area under the curve.

| Model | ELPD difference | SE difference | Weight | AUC | Classification accuracy | Accuracy (≥30) | Accuracy (<30) |
|---|---|---|---|---|---|---|---|
| Cumulative number of exposures, variant and age | 0.000 | 0.000 | 0.154 | 0.914 | 0.850 | 0.893 | 0.712 |
| Vaccination status, variant and age | –1.817 | 7.807 | 0.000 | 0.913 | 0.849 | 0.890 | 0.720 |
| Vaccination status and age | –6.061 | 8.899 | 0.134 | 0.913 | 0.849 | 0.888 | 0.724 |
| Cumulative number of exposures and age | –7.900 | 4.710 | 0.007 | 0.913 | 0.849 | 0.893 | 0.712 |
| Days since previous exposure, variant and age | –8.046 | 8.929 | 0.288 | 0.913 | 0.849 | 0.890 | 0.721 |
| Days since previous exposure and age | –20.190 | 9.360 | 0.000 | 0.912 | 0.850 | 0.890 | 0.725 |
| Variant and age | –22.457 | 8.628 | 0.081 | 0.912 | 0.850 | 0.888 | 0.728 |
| Vaccination status and variant | –59.936 | 14.986 | 0.162 | 0.909 | 0.847 | 0.888 | 0.717 |
| Cumulative number of exposures and variant | –60.139 | 12.098 | 0.007 | 0.910 | 0.846 | 0.891 | 0.703 |
| Age | –63.495 | 12.834 | 0.052 | 0.910 | 0.849 | 0.882 | 0.743 |
| Cumulative number of exposures | –65.893 | 12.851 | 0.114 | 0.910 | 0.846 | 0.891 | 0.707 |
| Vaccination status | –67.520 | 15.418 | 0.000 | 0.909 | 0.847 | 0.888 | 0.717 |
| Days since previous exposure and variant | –82.798 | 15.986 | 0.000 | 0.909 | 0.846 | 0.888 | 0.714 |
| Days since previous exposure | –93.537 | 16.058 | 0.000 | 0.909 | 0.846 | 0.887 | 0.718 |
| Variant | –107.362 | 15.800 | 0.000 | 0.907 | 0.845 | 0.883 | 0.723 |
| Baseline | –148.350 | 18.608 | 0.000 | 0.905 | 0.842 | 0.880 | 0.722 |

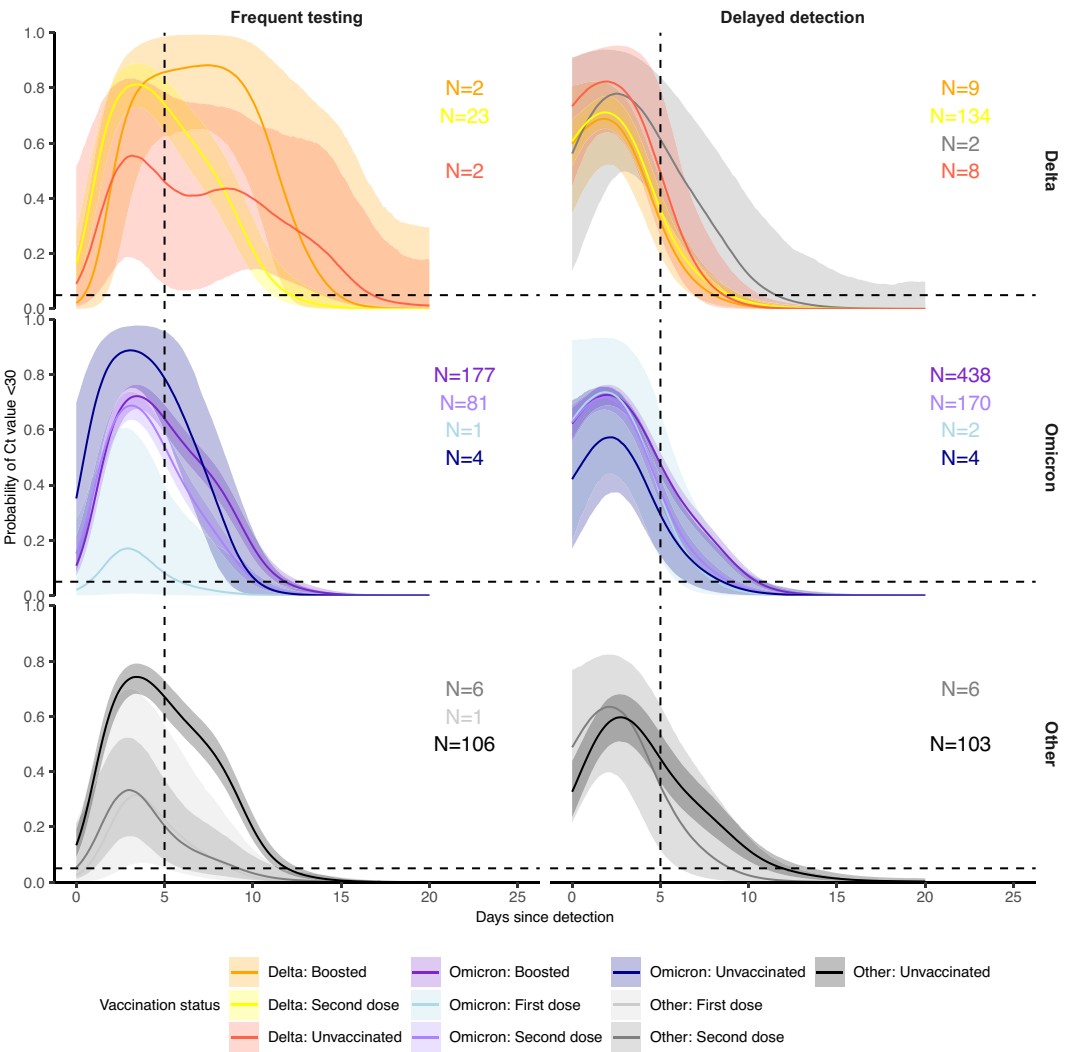

**Appendix 1—figure 9.** Proportion of infections with Ct <30 on each day post detection by confirmed or suspected variant, vaccination status and detection group, conditional on being in the <30 years age group. Solid colored lines and shaded ribbons are posterior estimates from a generalized linear model predicting probability of Ct <30 as a function of days since detection and vaccination status, showing the posterior mean (solid line) and 95% credible intervals (shaded ribbon) of each conditional effect. Dashed horizontal and vertical lines show 5% probability and day 5 post detection respectively. Labels show sample size within each group. Note that data is unavailable for some variant and vaccination status combinations.

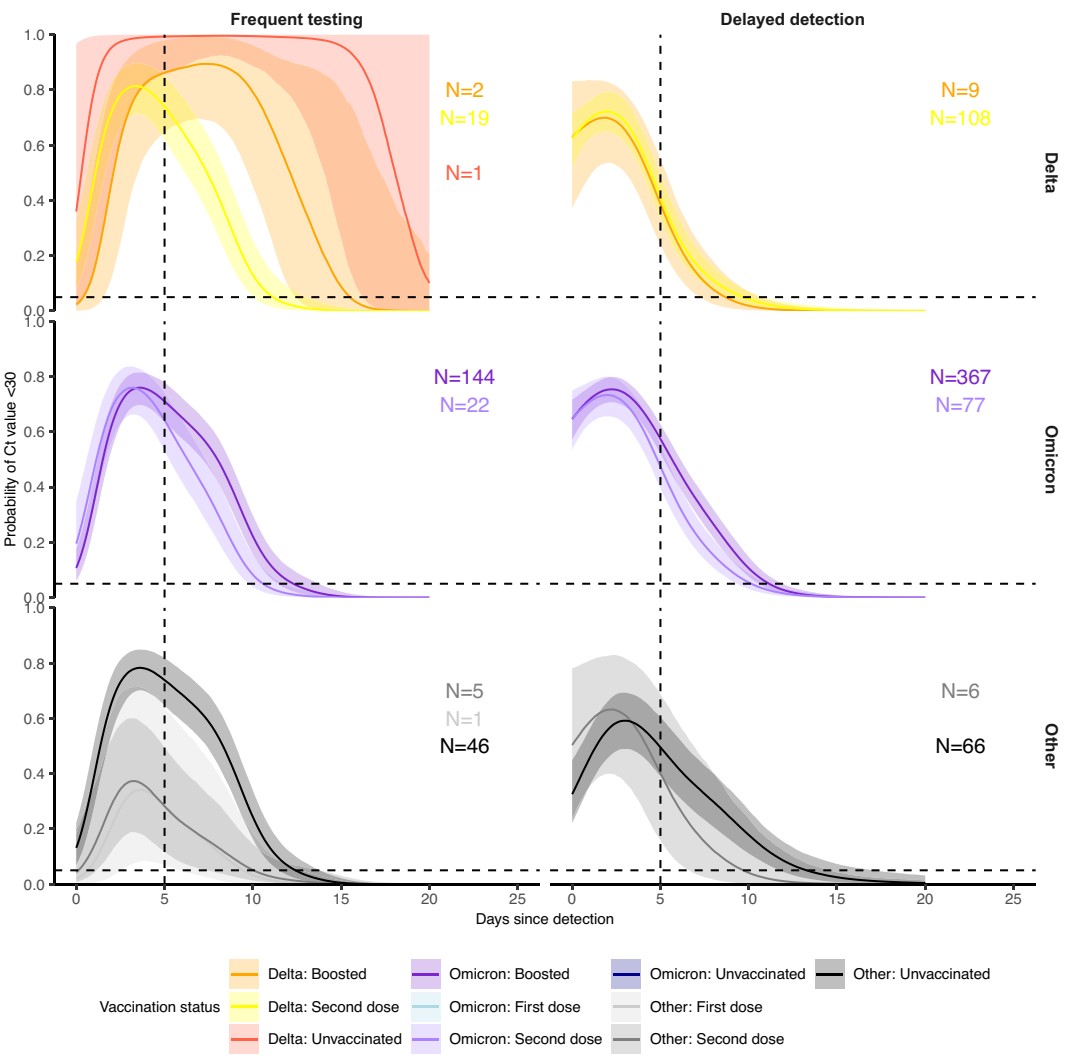

**Appendix 1—figure 10.** Identical to *Appendix 1—figure 9*, but after excluding data from all players.

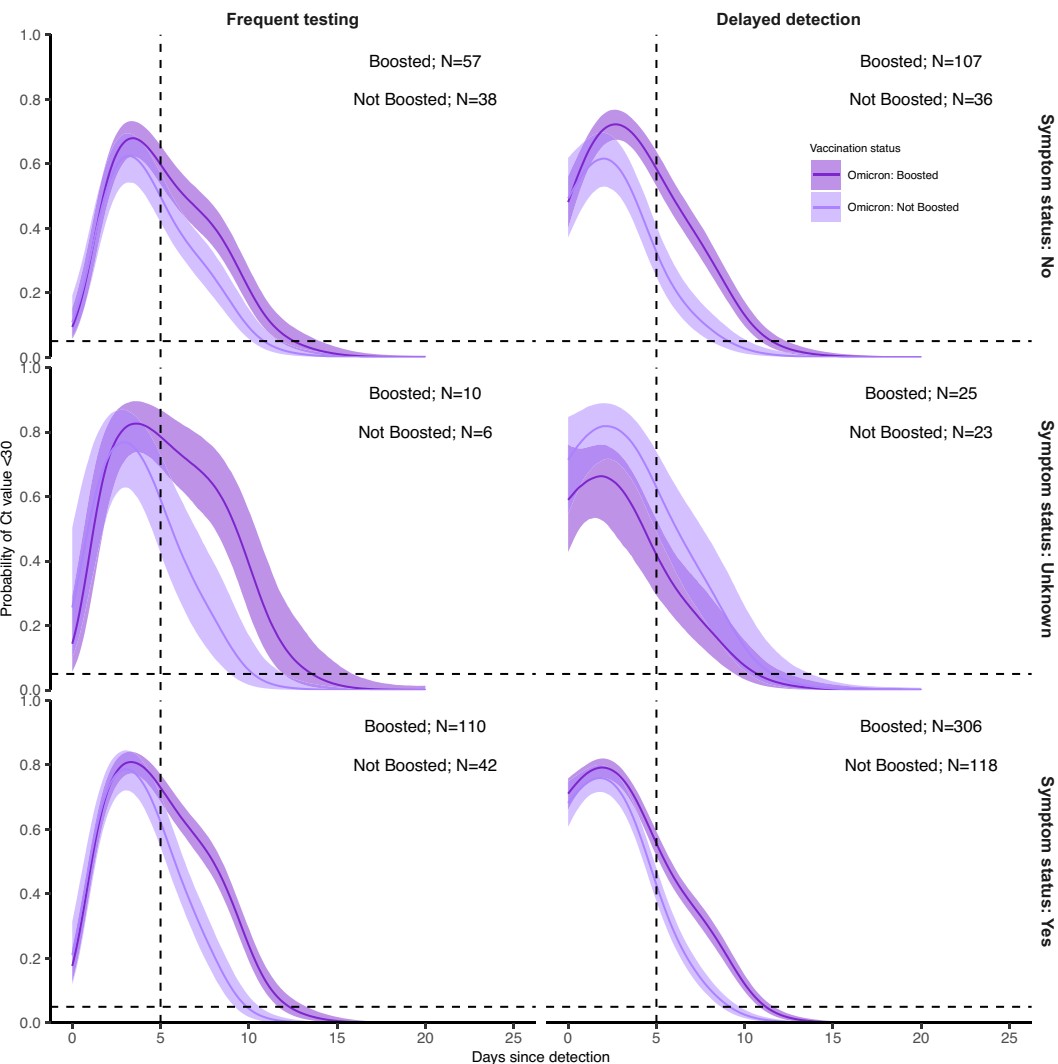

**Appendix 1—figure 11.** Proportion of Omicron BA.1 infections, stratified by symptom status, with Ct <30 on each day post detection by booster status and detection group. Solid colored lines and shaded ribbons are posterior estimates from a generalized linear model predicting probability of Ct <30 as a function of days since detection, variant (only BA.1 results shown) and vaccination status, showing the posterior mean (solid line) and 95% credible intervals (shaded ribbon) of each conditional effect. Dashed horizontal and vertical lines show 5% probability and day 5 post detection respectively. Note that age group is not included as an effect in this model.

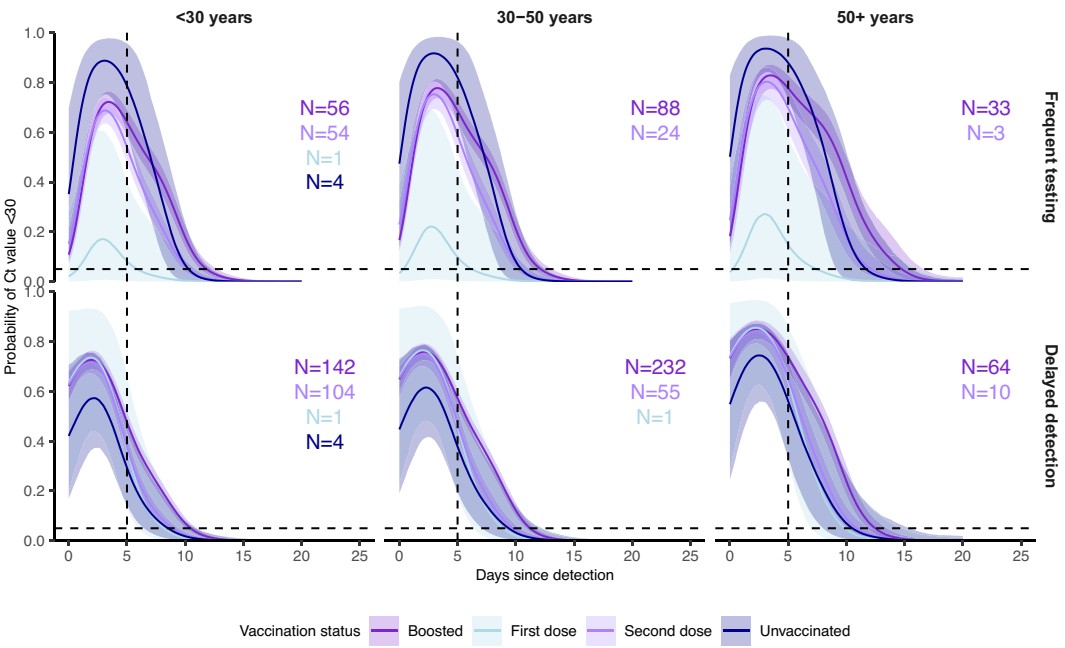

**Appendix 1—figure 12.** Proportion of Omicron BA.1 infections with Ct <30 on each day post detection stratified by vaccination status, age group and detection group. Solid colored lines and shaded ribbons are posterior estimates from a generalized linear model predicting probability of Ct <30 as a function of days since detection, vaccination status and variant, showing the posterior mean (solid line) and 95% credible intervals (shaded ribbon) of each conditional effect. Note that only Omicron BA.1 estimates are shown, though the model included data from Delta and pre-Delta infections. Dashed horizontal and vertical lines show 5% probability and day 5 post detection respectively.

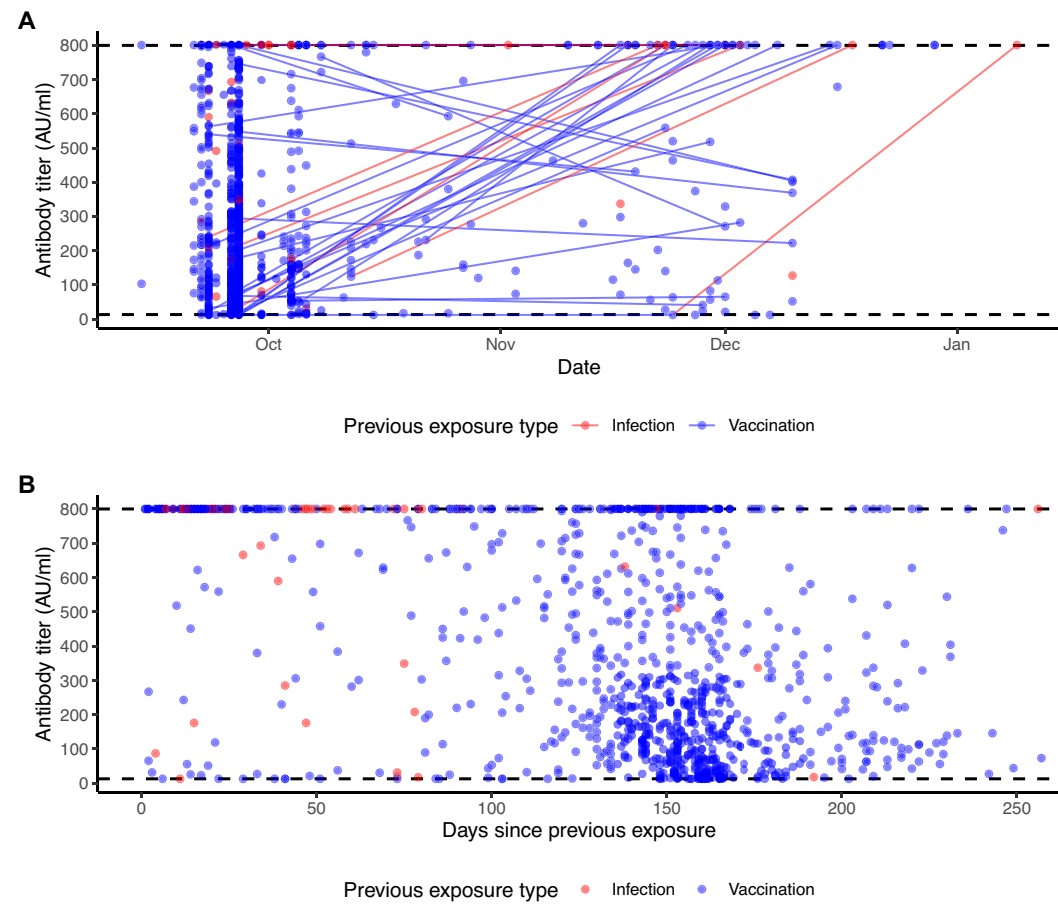

**Appendix 1—figure 13.** Individual antibody titers to the ancestral SARS-CoV-2 spike over time. (**A**) Measured antibody titers by date of sample collection. Lines show longitudinal samples from the same individual, colored by the most recent exposure at the time of sample collection. Lines going up therefore represent antibody boosting events, and lines going down represent waning. (**B**) Measured antibody titers by days since previous exposure at time of sample collection. Dashed lines show the limit of assay detection.

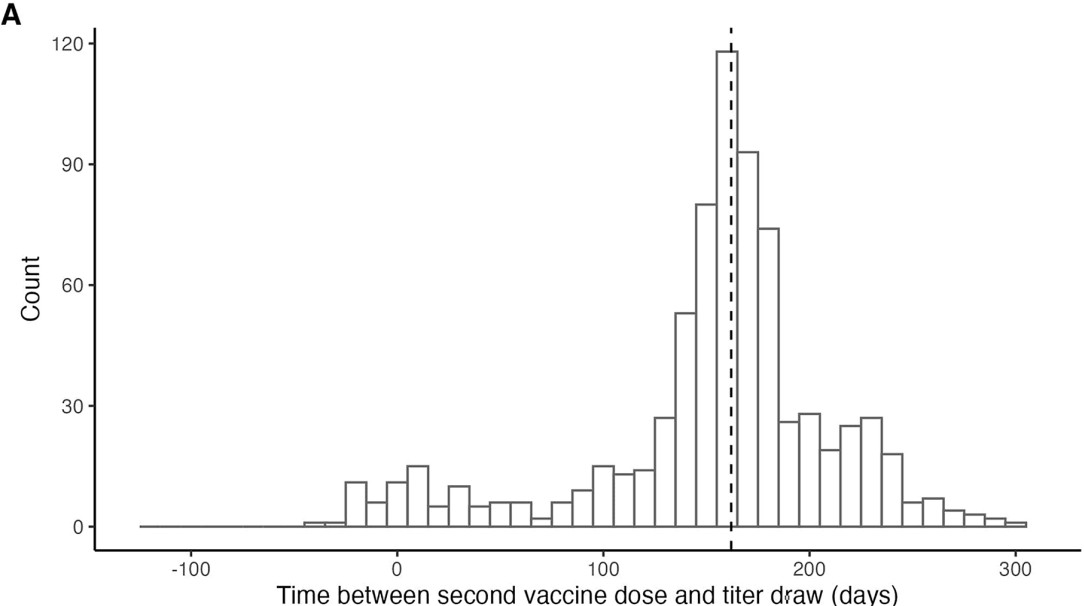

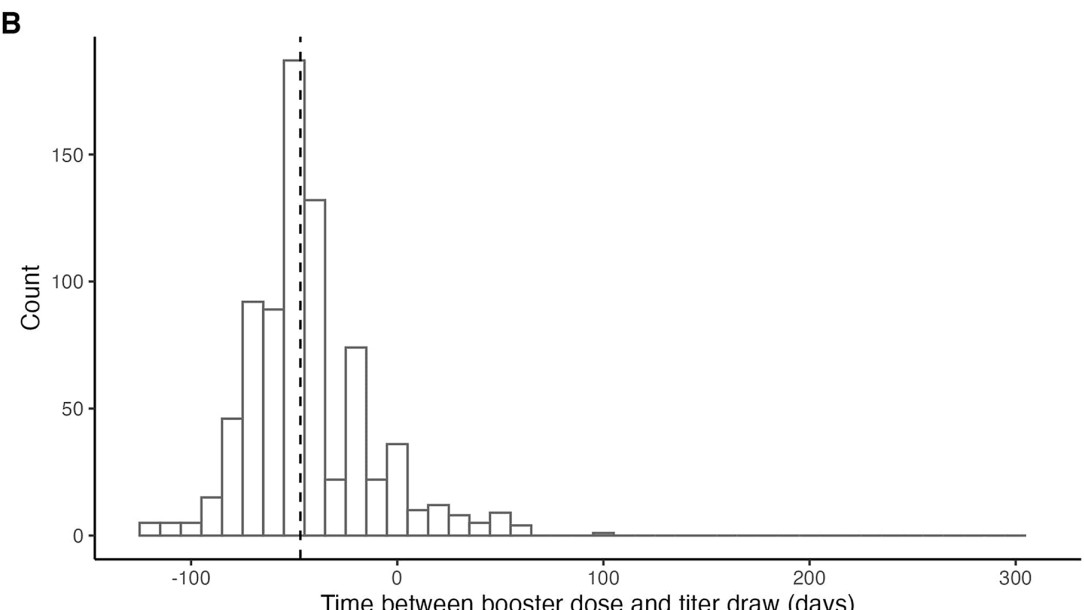

**Appendix 1—figure 14.** Histogram of time between (**A**) second vaccine dose and antibody titer measurement and (**B**) booster dose and antibody titer measurement. Dashed line marks the median lag (162 days). 1 individual was infected between receiving their second vaccine dose and having a titer measurement taken (Delta infection). 42 individuals were infected between having their titer measurement taken and receiving their booster vaccine dose (32 Delta; 9 unsequenced; 1 confirmed Omicron BA.1).

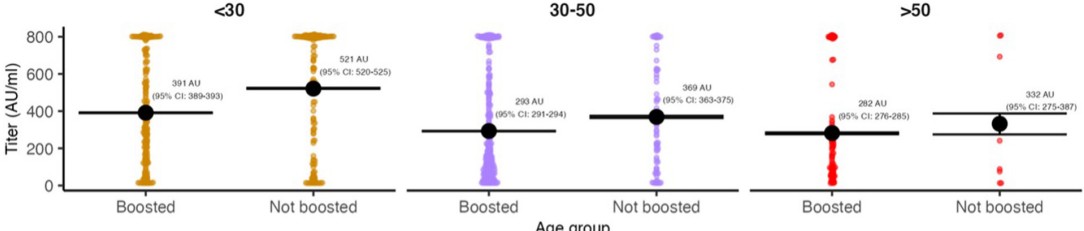

**Appendix 1—figure 15.** Distribution of antibody titers among Omicron BA.1-infected individuals (colored points) stratified by age group and vaccination status at the time of infection. Shown are mean titers (large black point) and bootstrapped 95% confidence intervals for the mean (black bars). Note that stratification is by infection and not individual, and that antibody titers were measured at a single point in time rather than near the time of infection. The Diasorin Trimeric Assay values are truncated between 13 and 800 AU/ml.

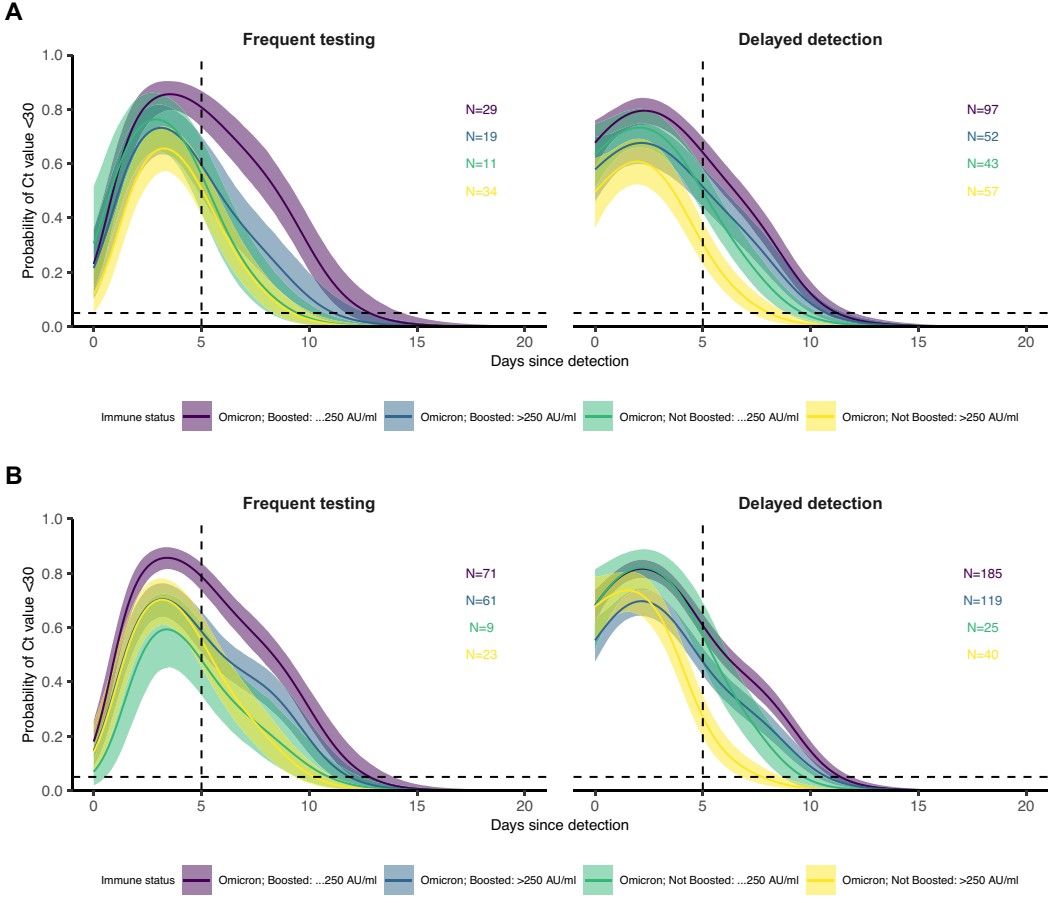

**Appendix 1—figure 16.** Proportion of BA.1 Omicron infections with Ct <30 on each day post detection stratified by the interaction of booster status at the time of infection and antibody titer group (note, not stratified by age group here due to small subgroup size). Shown are posterior estimates from a generalized linear model predicting probability of Ct <30 with a spline term on days since detection, conditional on titer/vaccination status category. Solid lines show posterior mean and shaded ribbons show 95% credible intervals of each conditional effect. (**A**) Model estimates including only individuals who had antibody titers measured between 100 and 200 days following a known previous infection of vaccination. (**B**) Model estimates including only infections which occurred between 60 and 90 days after an antibody titer measurement.

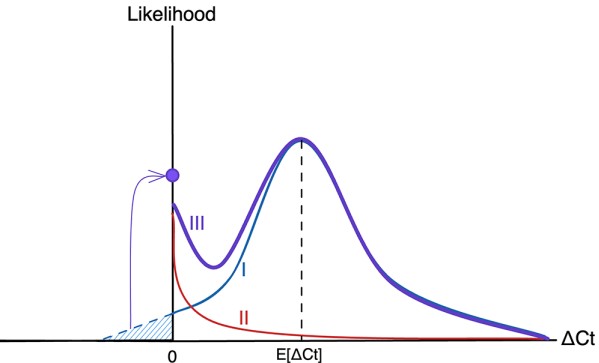

**Appendix 1—figure 17.** Schematic diagram of the likelihood function for viral kinetic inference. The plot depicts the likelihood as a function of ΔCt, the difference between the observed Ct value and the limit of detection, so that ΔCt = 0 (the origin) represents observations at the limit of detection, with viral load increasing toward the right-hand side of the plot. The likelihood function (III, purple) is made up of two fundamental components: the process likelihood (I, blue) and the false negative distribution (II, red). The main component of the process likelihood (**I**) is defined by a normal distribution with mean E[ΔCt], a function of the estimated viral kinetic parameters as defined by the viral kinetic model. Any mass of the process likelihood that extends below the limit of detection (blue hatched region) is instead added to a point probability mass at the origin, since viral loads below the limit of detection register at the limit of detection. The false negative distribution (II) is an exponential distribution with fixed rate to account for a small amount of noise near the limit of detection. Summing the process likelihood (**I**) and the false negative likelihood (**I**) using the mixing probability $\lambda$ (=1 - sensitivity) yields the overall likelihood (III).

**Appendix 1—table 5.** Posterior estimates of viral trajectory attributes by variant and vaccination status.

Estimates are posterior means with 95% credible intervals.

| Trajectory value | Variant/vaccination status | Estimate |
| --- | --- | --- |
| Peak viral load (Ct) | Other: Unvaccinated | 25.0 (24.2, 25.9) |
| | Delta: 1–2 doses | 22.4 (21.4, 23.5) |
| | Omicron: 1–2 doses | 25.6 (25.0, 26.2) |
| | Omicron: Boosted | 25.7 (25.4, 26.1) |
| Peak viral load ($\log_{10}$ copies/ml) | Other: Unvaccinated | 6.8 (6.6, 7.0) |
| | Delta: 1–2 doses | 7.5 (7.2, 7.8) |
| | Omicron: 1–2 doses | 6.6 (6.5, 6.8) |
| | Omicron: Boosted | 6.6 (6.5, 6.7) |
| Proliferation time (days) | Other: Unvaccinated | 3.5 (3.1, 3.9) |
| | Delta: 1–2 doses | 3.6 (3.1, 4.1) |
| | Omicron: 1–2 doses | 3.6 (3.3, 4.0) |
| | Omicron: Boosted | 4.0 (3.8, 4.3) |
| Clearance time (days) | Other: Unvaccinated | 9.9 (9.2, 10.6) |
| | Delta: 1–2 doses | 7.6 (7.0, 8.3) |
| | Omicron: 1–2 doses | 6.2 (5.8, 6.6) |
| | Omicron: Boosted | 8.4 (8.0, 8.7) |

**Appendix 1—table 6.** Posterior estimates of viral trajectory attributes by variant and antibody titer. Estimates are posterior means with 95% credible intervals.

| Trajectory attribute | Variant/immune status | Estimate |
|---|---|---|
| Peak viral load (Ct) | Other: Unexposed | 24.8 (24.0, 25.6) |
| | Delta: Exposed,≤250 AU | 22.1 (20.6, 23.4) |
| | Delta: Exposed,>250 AU | 24.4 (22.7, 26.1) |
| | Omicron: Exposed,≤250 AU | 25.2 (24.7, 25.6) |
| | Omicron: Exposed,>250 AU | 26.2 (25.7, 26.6) |
| Peak viral load (log$_{10}$ copies/ml) | Other: Unexposed | 6.9 (6.6, 7.1) |
| | Delta: Exposed,≤250 AU | 7.6 (7.3, 8.0) |
| | Delta: Exposed,>250 AU | 7.0 (6.5, 7.4) |
| | Omicron: Exposed,≤250 AU | 6.8 (6.7, 6.9) |
| | Omicron: Exposed,>250 AU | 6.5 (6.4, 6.6) |
| Proliferation time (days) | Other: Unexposed | 3.5 (3.2, 3.9) |
| | Delta: Exposed,≤250 AU | 3.6 (2.8, 4.5) |
| | Delta: Exposed,>250 AU | 4.2 (3.3, 5.3) |
| | Omicron: Exposed,≤250 AU | 3.9 (3.6, 4.2) |
| | Omicron: Exposed,>250 AU | 3.7 (3.5, 4.0) |
| Clearance time (days) | Other: Unexposed | 9.8 (9.1, 10.6) |
| | Delta: Exposed,≤250 AU | 7.7 (6.9, 8.7) |
| | Delta: Exposed,>250 AU | 7.6 (6.4, 8.8) |
| | Omicron: Exposed,≤250 AU | 8.4 (8.0, 8.8) |
| | Omicron: Exposed,>250 AU | 6.9 (6.6, 7.3) |

**Appendix 1—table 7.** Posterior estimates of Omicron BA.1 viral trajectory attributes by symptom and vaccination status.

Estimates are posterior means with 95% credible intervals.

| Trajectory attribute | Variant/immune status | Estimate |
|---|---|---|
| Peak viral load (Ct) | Omicron: 1–2 doses, no symptoms | 26.7 (25.7, 27.7) |
| | Omicron: 1–2 doses, symptoms | 25.3 (24.5, 26.1) |
| | Omicron: boosted, no symptoms | 26.3 (25.7, 27) |
| | Omicron: boosted, symptoms | 25.4 (25, 25.8) |
| Peak viral load (log$_{10}$ copies/ml) | Omicron: 1–2 doses, no symptoms | 6.3 (6.1, 6.6) |
| | Omicron: 1–2 doses, symptoms | 6.7 (6.5, 6.9) |
| | Omicron: boosted, no symptoms | 6.4 (6.3, 6.6) |
| | Omicron: boosted, symptoms | 6.7 (6.6, 6.8) |
| Proliferation time (days) | Omicron: 1–2 doses, no symptoms | 3.7 (3.2, 4.2) |
| | Omicron: 1–2 doses, symptoms | 3.7 (3.2, 4.2) |
| | Omicron: boosted, no symptoms | 4.3 (3.9, 4.8) |
| | Omicron: boosted, symptoms | 3.8 (3.5, 4.1) |
| Clearance time (days) | Omicron: 1–2 doses, no symptoms | 5.9 (5.2, 6.6) |
| | Omicron: 1–2 doses, symptoms | 6.3 (5.8, 6.8) |
| | Omicron: boosted, no symptoms | 7.4 (6.8, 8.0) |
| | Omicron: boosted, symptoms | 8.7 (8.3, 9.1) |

**Appendix 1—table 8.** Posterior estimates of Omicron BA.1 viral trajectory attributes by symptom and antibody titer.

Estimates are posterior means with 95% credible intervals.

| Trajectory attribute | Variant/immune status | Estimate |
|---|---|---|
| Peak viral load (Ct) | Omicron: Low titer, no symptoms | 26.1 (25.2, 26.9) |
| | Omicron: Low titer, symptoms | 24.8 (24.3, 25.4) |
| | Omicron: High titer, no symptoms | 26.7 (25.9, 27.5) |
| | Omicron: High titer, symptoms | 25.9 (25.4, 26.5) |
| Peak viral load ($\log_{10}$ copies/ml) | Omicron: Low titer, no symptoms | 6.5 (6.3, 6.8) |
| | Omicron: Low titer, symptoms | 6.9 (6.7, 7.0) |
| | Omicron: High titer, no symptoms | 6.3 (6.1, 6.6) |
| | Omicron: High titer, symptoms | 6.6 (6.4, 6.7) |
| Proliferation time (days) | Omicron: Low titer, no symptoms | 4.3 (3.8, 4.9) |
| | Omicron: Low titer, symptoms | 3.6 (3.3, 4.0) |
| | Omicron: High titer, no symptoms | 3.8 (3.4, 4.3) |
| | Omicron: High titer, symptoms | 3.8 (3.4, 4.2) |
| Clearance time (days) | Omicron: Low titer, no symptoms | 7.4 (6.7, 8.1) |
| | Omicron: Low titer, symptoms | 8.7 (8.2, 9.3) |
| | Omicron: High titer, no symptoms | 6.4 (5.8, 7.1) |
| | Omicron: High titer, symptoms | 7.1 (6.6, 7.5) |

**Appendix 1—table 9.** Posterior estimates of Omicron BA.1 viral RNA clearance times by age and vaccination status.

Estimates are posterior means with 95% credible intervals. Low titer is ≤250 AU, high titer is >250 AU.

| Trajectory attribute | Variant/immune status | Estimate |
|---|---|---|
| Peak viral load (Ct) | Omicron:<30, unboosted | 25.6 (25, 26.3) |
| | Omicron:<30, boosted | 25.8 (25.1, 26.4) |
| | Omicron: 30–50, unboosted | 25.3 (24.2, 26.5) |
| | Omicron: 30–50, boosted | 25.7 (25.2, 26.2) |
| | Omicron: 50+, unboosted | 25.5 (22.8, 27.9) |
| | Omicron: 50+, boosted | 25.5 (24.6, 26.3) |
| Peak viral load ($\log_{10}$ copies/ml) | Omicron:<30, unboosted | 6.6 (6.4, 6.8) |
| | Omicron:<30, boosted | 6.6 (6.4, 6.8) |
| | Omicron: 30–50, unboosted | 6.7 (6.4, 7) |
| | Omicron: 30–50, boosted | 6.6 (6.5, 6.7) |
| | Omicron: 50+, unboosted | 6.7 (6, 7.4) |
| | Omicron: 50+, boosted | 6.7 (6.5, 6.9) |
| Proliferation time (days) | Omicron:<30, unboosted | 3.7 (3.4, 4.1) |
| | Omicron:<30, boosted | 4.2 (3.8, 4.7) |
| | Omicron: 30–50, unboosted | 3.6 (3, 4.4) |
| | Omicron: 30–50, boosted | 4 (3.6, 4.3) |
| | Omicron: 50+, unboosted | 4.4 (3, 6.3) |

*Appendix 1—table 9 Continued on next page*

*Appendix 1—table 9 Continued*

| Trajectory attribute | Variant/immune status | Estimate |
|---|---|---|
| | Omicron: 50+, boosted | 3.9 (3.4, 4.4) |
| Clearance time (days) | Omicron:<30, unboosted | 6 (5.6, 6.5) |
| | Omicron:<30, boosted | 7.2 (6.7, 7.7) |
| | Omicron: 30–50, unboosted | 6.4 (5.7, 7.2) |
| | Omicron: 30–50, boosted | 8.6 (8.1, 9.1) |
| | Omicron: 50+, unboosted | 8.7 (6.6, 11.2) |
| | Omicron: 50+, boosted | 9.6 (8.8, 10.6) |

**Appendix 1—table 10.** Posterior estimates of Omicron BA.1 viral RNA clearance times by age and titer.
Estimates are posterior means with 95% credible intervals. Low titer is ≤250 AU, high titer is >250 AU.

| Trajectory attribute | Variant/immune status | Estimate |
|---|---|---|
| Peak viral load (Ct) | Omicron:<30, low titer | 24.8 (24, 25.5) |
| | Omicron:<30, high titer | 26.2 (25.6, 26.8) |
| | Omicron: 30–50, low titer | 25.4 (24.8, 26) |
| | Omicron: 30–50, high titer | 26.1 (25.4, 26.8) |
| | Omicron: 50+, low titer | 24.9 (23.8, 25.9) |
| | Omicron: 50+, high titer | 26.3 (24.8, 27.6) |
| Peak viral load ($\log_{10}$ copies/ml) | Omicron:<30, low titer | 6.9 (6.7, 7.1) |
| | Omicron:<30, high titer | 6.5 (6.3, 6.7) |
| | Omicron: 30–50, low titer | 6.7 (6.5, 6.9) |
| | Omicron: 30–50, high titer | 6.5 (6.3, 6.7) |
| | Omicron: 50+, low titer | 6.8 (6.6, 7.1) |
| | Omicron: 50+, high titer | 6.5 (6.1, 6.9) |
| Proliferation time (days) | Omicron:<30, low titer | 3.9 (3.4, 4.3) |
| | Omicron:<30, high titer | 4 (3.6, 4.4) |
| | Omicron: 30–50, low titer | 4 (3.6, 4.4) |
| | Omicron: 30–50, high titer | 3.6 (3.2, 4) |
| | Omicron: 50+, low titer | 3.8 (3.2, 4.5) |
| | Omicron: 50+, high titer | 3.7 (2.9, 4.7) |
| Clearance time (days) | Omicron:<30, low titer | 7.2 (6.7, 7.8) |
| | Omicron:<30, high titer | 6.2 (5.8, 6.7) |
| | Omicron: 30–50, low titer | 8.7 (8.1, 9.3) |
| | Omicron: 30–50, high titer | 7.4 (6.8, 8) |
| | Omicron: 50+, low titer | 9.9 (8.7, 11.1) |
| | Omicron: 50+, high titer | 9.5 (8, 11.1) |

