## [Editor Report]

This manuscript provides a valuable and policy-relevant contribution to our understanding of SARS-CoV-2 viral kinetics in the Omicron era. The authors exploit a rich and unique dataset from the National Basketball Association to describe post-infection viral kinetics, including rebounds, and to explore evidence for differential kinetics by immune history and demographics. The authors show (as others have) that most people remain with high viral loads 5 days post positive test and that older individuals and those who were boosted (but had a poor initial antibody response to the primary vaccine series) were more likely to remain with high viral loads longer after an Omicron infection.

---

## [Decision Letter]

**Decision letter after peer review:**

Thank you for submitting your article "Quantifying the impact of immune history and variant on SARS-CoV-2 viral kinetics and infection rebound: a retrospective cohort study" for consideration by *eLife*. Your article has been reviewed by 2 peer reviewers, and the evaluation has been overseen by a Reviewing Editor and Neil Ferguson as the Senior Editor. The following individual involved in review of your submission has agreed to reveal their identity: Andrew Azman (Reviewer #2).

Essential revisions:

1) Please add a description (table?) of the demographics of the participants (e.g. age, sex, co-morbidities), stratified by important factors (i.e. vaccination status, exposure history, antibody titer class).

2) It would be helpful to add the models with age effects (uni and multivariate) to model performance comparison in Tables S3/4.

3) While the analyses stratified by antibody levels are interesting, the differences in kinetics that remain, even after stratifying, are also intriguing. It might not be possible to present an analysis stratified by antibody level and age, but it would be useful to discuss to what extent age might explain these differences. Presenting a figure (like Figure 2b) stratified by age could also be useful. Authors should also discuss alternative hypotheses for differences in kinetics between boosted and unboosted individuals.

4) Please clarify whether any of the participants took antivirals.

*Reviewer #1 (Recommendations for the authors):*

Although the effect of age as a confounder is accounted for and discussed, I would have found it helpful to see it considered from the beginning as a potentially important factor and tested along with a cumulative number of exposures, lineage, days since previous exposure and vaccination status. It would be interesting to see a model that in addition to the baseline spline also accounts for a single predictor of age in the results presented in Tables S3 and S4. In addition, I would consider showing some of the results for age and Omicron in the main text.

I found the methods and results for the logistic regression a bit difficult to follow at some points, with most of the results presented in the supplementary information, implying going back and forth between the two files.

– It would be helpful to add a list (or Table) with all possible factors (predictors) considered and the values (or categories of values) they can take (e.g. age: <30, 30-50, >50).

– How is the effect of each factor modelled? It is written in the methods section that "additional logistic regression models, adding additional spline terms capturing the effect of vaccination status, cumulative number of previous exposures or days since previous exposure, and/or lineage with days since detection". Does it mean an additional spline is added for each possible category of each variable? If so, how is "days since previous exposure" treated? I understand that for the models that include more than one predictor, those are always considered to interact – is that true? Please clarify.

– The caption of Figure S9 needs to be rewritten and more precise. First, panel (A) only shows the results for a subset of the data (BA.1 infected and boosted individuals). Therefore, I would avoid saying "conditioning on vaccination status and lineage", which suggests that the whole dataset has been used to fit the model. Similar for panel (B). Second, it would be helpful to make clearer which model has been used for the results in panel (A) and which one for the results in panel (B). Now it is only said at the end of the caption. Third, the model for panel (B) is described as including an interaction between days since detection with vaccination status and variant. However, as far as I understand, only Omicron infections are considered here, which means that variant is in practice not used as a covariate, and therefore, the model only has 2 splines (one for boosted and one for non-boosted individuals).

– In most figures, results are presented for the frequent testing group and the delayed detection group. Are the logistic models fitted jointly to the whole dataset or independently to each of the two groups? I understand that the latter. In that case, I think it is not correct to say "stratified". Instead, I would just say that the model has been independently fitted to X and Y.

Methods section, Viral kinetic model (lines 589-604). (Again) when the authors say they "stratify" the model by a certain variable, what do they mean? Do they just simply fit the model independently to each subset of data, or something else? Please clarify.

Methods section, Incidence of rebounds (lines 145-169). Although it might seem obvious, I think it would be good to make clear they talk about consecutive days in the definition of a rebound.

For consistency throughout the manuscript, I would suggest using either: (1) variant or lineage, (2) vaccination status and vaccination history, (3) Omicron or BA.1.

I have not seen any statement about the availability of the data. Please clarify.

*Reviewer #2 (Recommendations for the authors):*

– It seems like the authors suggest that no one took antivirals after infection in this cohort but I don't see this explicitly stated here. Do you have data on this or it is assumed?

– Throughout reading this I found myself wanting to see a better description of the characteristics of the participants (eg, age, sex, co-morbidities [which there probably aren't many of]), ideally stratified by important factors like vaccination and exposure history and titer class (high/low).

– In the models predicting Pr(Ct<30), it is not clear why cross-validation is not being used for AUC and classification accuracy measures. Not an issue with EPLD shown in the supplemental table.

– It would be helpful to add the models with age effects to model performance comparison in Tables S3/4.

– Figure 2C – 95%CrI shown horizontally seems strange. Are these not meant to be aligned with the y-axis?

– Figure S11 is hard to read and I do not believe the legend and caption fully describe what is going on. Would be helpful to update this.

– line 131 – can you add the n to this?

– Code on github is easy to use and pretty well documented, thank you!

---

## [Author Response]

Essential revisions:1) Please add a description (table?) of the demographics of the participants (e.g. age, sex, co-morbidities), stratified by important factors (i.e. vaccination status, exposure history, antibody titer class).

We have added an additional table (Table 1). Table 1 gives demographic characteristics attached to each infection (note some individuals have multiple infections) as well as the factor levels used in the modeling analyses. Unfortunately, we are unable to add additional granularity to Table 1 regarding age, sex, co-morbidities or role within the NBA due to the sensitive nature of these data and the data use agreement.

2) It would be helpful to add the models with age effects (uni and multivariate) to model performance comparison in Tables S3/4.

We have included all of the additional models including age effects in the Tables S3/4 as suggested. We note that this has doubled the number of presented models, as we consider all of the previous 8 models with an additional spline term for age group.

3) While the analyses stratified by antibody levels are interesting, the differences in kinetics that remain, even after stratifying, are also intriguing. It might not be possible to present an analysis stratified by antibody level and age, but it would be useful to discuss to what extent age might explain these differences. Presenting a figure (like Figure 2b) stratified by age could also be useful. Authors should also discuss alternative hypotheses for differences in kinetics between boosted and unboosted individuals.

Figure S15 presents the same analyses as Figure 2 but with an additional stratification by age group. As you allude to, one of the limitations with this additional level of stratification is that the group sizes for some of the age/vaccination status/antibody titer groups are small (or indeed 0), particularly in the 50+ age group. Nonetheless, this stratification shows that (a) the pattern of lower titers among boosted, BA.1-infected individuals remains within each age group and (b) the pattern for longer duration of Ct<30 in boosted, low titer individuals remains for the <30 and 30-50 age groups. We have rearranged the figures and rewritten much of the Results section to better incorporate the effect of age into the narrative:

1. Previous Figure S9 has now become main text Figure 2, showing the marginal effect of age group and vaccination status on duration Ct<30 from Omicron BA.1 infections.

2. The results from Figure S15 have been combined with Figure 2C to create a new main text Figure 3. This gives the results from the same model as was previously shown in Figure 2B, but now including an additional stratification by age group as suggested by the reviewer. We note that although we only show the Omicron BA.1 results (as they are the only results of interest here), this model did include data from other lineages in the fitting.

If the editor and reviewers have suggestions for alternative hypotheses for the differences in the kinetics between boosted and unboosted individuals, we would welcome them.

4) Please clarify whether any of the participants took antivirals.

They did not. We now state this in the first paragraph of the “Data” section.

Reviewer #1 (Recommendations for the authors):Although the effect of age as a confounder is accounted for and discussed, I would have found it helpful to see it considered from the beginning as a potentially important factor and tested along with a cumulative number of exposures, lineage, days since previous exposure and vaccination status. It would be interesting to see a model that in addition to the baseline spline also accounts for a single predictor of age in the results presented in Tables S3 and S4. In addition, I would consider showing some of the results for age and Omicron in the main text.

We now refer to the models including age from the start of the logistic regression results. Rather than referring to this as a sensitivity analysis, we now consider all of the models including age as part of the initial model comparison analysis. We have replaced main text Figure 2 with a figure describing the model including vaccination status and age group (previously Figure S9).

I found the methods and results for the logistic regression a bit difficult to follow at some points, with most of the results presented in the supplementary information, implying going back and forth between the two files.

Apologies for this. We have tried to reframe the start of the Results section to highlight that we are building successively more complex models from an initial baseline considering only days since detection as a predictor. We hope that the reader will interpret this as an exhaustive comparison of potential classification models rather than walking through the results of each model fit.

– It would be helpful to add a list (or Table) with all possible factors (predictors) considered and the values (or categories of values) they can take (e.g. age: <30, 30-50, >50).

We have added new a Table 1 to the main text to summarize the data and show the possible categories for each variable.

– How is the effect of each factor modelled? It is written in the methods section that "additional logistic regression models, adding additional spline terms capturing the effect of vaccination status, cumulative number of previous exposures or days since previous exposure, and/or lineage with days since detection". Does it mean an additional spline is added for each possible category of each variable? If so, how is "days since previous exposure" treated? I understand that for the models that include more than one predictor, those are always considered to interact – is that true? Please clarify.

We have added text to the methods to clarify how the models consider effects for each predictor. In brief, consider the baseline model as a single smoothing spline on days since detection as a predictor. Each model then adds an intercept and spline term over days since detection (on top of this baseline model) for each possible category. In models including variant, we consider the interaction of variant and exposure history/vaccination status/days since exposure group. We do not add an interaction between age group and any other variable – age group is always an additive effect. These additional spline terms therefore attempt to explain variation not explained by days since detection alone. Regarding the “days since previous exposure” variable, we discretized each exposure into “Naïve”, “<1 month”, “1-3 months” or “>3 months” relative to the most recent prior exposure. This is now described in the methods.

– The caption of Figure S9 needs to be rewritten and more precise. First, panel (A) only shows the results for a subset of the data (BA.1 infected and boosted individuals). Therefore, I would avoid saying "conditioning on vaccination status and lineage", which suggests that the whole dataset has been used to fit the model. Similar for panel (B). Second, it would be helpful to make clearer which model has been used for the results in panel (A) and which one for the results in panel (B). Now it is only said at the end of the caption. Third, the model for panel (B) is described as including an interaction between days since detection with vaccination status and variant. However, as far as I understand, only Omicron infections are considered here, which means that variant is in practice not used as a covariate, and therefore, the model only has 2 splines (one for boosted and one for non-boosted individuals).

Apologies for the confusion here – all results in this figure are from the same model. This figure shows the conditional effect of age or vaccination status from the same model which includes spline terms for age group and the interaction of variant with vaccination status. The splines shown are the conditional effects of (A) age group after conditioning on being a BA.1 infection and boosted, or (B) vaccination status after conditioning on being a BA.1 infection and age group <30 years. Thus, it is correct that the whole dataset was used to fit this model, as the figure shows conditional effects. We have updated the figure caption to clarify these points.

– In most figures, results are presented for the frequent testing group and the delayed detection group. Are the logistic models fitted jointly to the whole dataset or independently to each of the two groups? I understand that the latter. In that case, I think it is not correct to say "stratified". Instead, I would just say that the model has been independently fitted to X and Y.

We have clarified that in all instances, the logistic regression models are fit independently to the two groups.

Methods section, Viral kinetic model (lines 589-604). (Again) when the authors say they "stratify" the model by a certain variable, what do they mean? Do they just simply fit the model independently to each subset of data, or something else? Please clarify.

To stratify the viral kinetic model, we chose a reference category and fitted independent additive random effects for the other categories. We clarify this in the new Methods section.

Methods section, Incidence of rebounds (lines 145-169). Although it might seem obvious, I think it would be good to make clear they talk about consecutive days in the definition of a rebound.

Thank you. We have clarified this point.

For consistency throughout the manuscript, I would suggest using either: (1) variant or lineage, (2) vaccination status and vaccination history, (3) Omicron or BA.1.

We have changed all instances of “vaccination history” to “vaccination status”. Regarding variant vs. lineage and Omicron vs. BA.1, we have checked for consistency but note that in some instances the distinction between the terms is important. For example, although we refer to Omicron BA.1 infections throughout much of the text, there are some instances where variant or lineage are appropriate, such as “pre-Omicron variants”, “BA.1 lineages BA.1.1529 and BA.1.1” or “non-Omicron variant”, which must remain as such to retain their intended meaning.

I have not seen any statement about the availability of the data. Please clarify.

The Data Availability statement contains a link to the data and code used for this study.

Reviewer #2 (Recommendations for the authors):– It seems like the authors suggest that no one took antivirals after infection in this cohort but I don't see this explicitly stated here. Do you have data on this or it is assumed?

We confirm that none of the study participants took antivirals. We now state this in the first paragraph of the revised Data section.

– Throughout reading this I found myself wanting to see a better description of the characteristics of the participants (eg, age, sex, co-morbidities [which there probably aren't many of]), ideally stratified by important factors like vaccination and exposure history and titer class (high/low).

We have added this information in Table 1 of the revised manuscript. We note that due to the sensitive nature of the data we are unable to provide more granularity on characteristics such as age, role within the NBA, co-morbidities etc.

– In the models predicting Pr(Ct<30), it is not clear why cross-validation is not being used for AUC and classification accuracy measures. Not an issue with EPLD shown in the supplemental table.

As suggested, we have replaced the AUC and classification accuracy measures with the results from the k-folds cross-validation analyses rather than using the within-sample measures. We note that although cross-validation results are generally regarded to be more reliable, this change has not affected any of the results.

– It would be helpful to add the models with age effects to model performance comparison in Tables S3/4.

We have added the age effects models into the comparison table as suggested. Models including age perform better than models without in terms of predictive accuracy, but they do not have substantially better classification accuracies. We refer to this in the Results section.

– Figure 2C – 95%CrI shown horizontally seems strange. Are these not meant to be aligned with the y-axis?

These CIs were correctly aligned with the y-axis, but were confusing given how narrow the intervals are. We elected to remove the CIs from the plot and simply show the mean as a horizontal line. We left the CIs as labels. We describe this in the figure caption.

– Figure S11 is hard to read and I do not believe the legend and caption fully describe what is going on. Would be helpful to update this.

We updated the figure caption to better describe this figure. Although there is a lot going on, we intended for this figure to be an exhaustive representation of all of the events of interest explored in the dataset over time.

– line 131 – can you add the n to this?

Added as suggested.

– Code on github is easy to use and pretty well documented, thank you!

You are very welcome.